

# Renormalization group and effective potential: A simple non-perturbative approach

**José Gaite**

Applied Physics Dept., ETSIAE, Universidad Politécnica de Madrid,
E-28040 Madrid, Spain

## Abstract

We develop a simple non-perturbative approach to the calculation of a field theory effective potential that is based on the Wilson or exact renormalization group. Our approach follows Shepard *et al*'s idea [Phys. Rev. D51, 7017 (1995)] of converting the exact renormalization group into a self-consistent renormalization method. It yields a simple second order differential equation for the effective potential. The equation can be solved and its solution is compared with other non-perturbative results and with results of perturbation theory. In three dimensions, we are led to study the sextic field theory ($\lambda\phi^4 + g\phi^6$). We work out this theory at two-loop perturbative order and find the non-perturbative approach to be superior.

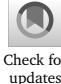

# 1 Introduction

Wilson is the main author of the modern theory of the renormalization group as well as of its formulation for statistical systems as a transformation of the full probability distribution, defined by an unbounded set of parameters [1]. The exact renormalization group (ERG) is the group (or semigroup) formed by such transformations. The ERG equation is a functional differential equation that admits several formulations [1–9]. The solution of the ERG equation should provide the exact infrared-cutoff-dependent generator of the Green functions and, hence, a full solution of the field theory in question. Of course, the exact renormalization group is not the only approach that establishes functional equations that provide the exact Green functions. The well-known Dyson-Schwinger equations constitute another set of functional equations with the same goal [10, 11]. These equations involve no cutoff, in principle, but actually need ultraviolet regularization. In fact, the presence of a running infrared cutoff in the ERG equation is an asset [7].

At any rate, neither the Dyson-Schwinger equations nor the ERG equation are easy to handle, since they involve functionals. Naturally, functions are easier to handle, and a field theory function that contains considerable information is the *effective potential* [10]. The ERG equation admits an approximation in terms of a *local potential* which plays the role of effective potential [4, 5, 12–15]. Of course, the expression of the full action or Hamiltonian as a series expansion in field derivatives is old and is the origin of the Landau-Ginzburg Lagrangian model of critical phenomena [16]. The ERG equation in the local potential approximation is a partial differential equation with derivatives respect to the renormalization group parameter and the field. At a fixed point, it becomes a nonlinear ordinary differential equation, which is nonetheless too hard to solve analytically. However, this equation has allowed to find, in three dimensions, the Wilson-Fisher fixed point and to compute quite accurate critical exponents [8].

The solution of the ERG equation in the local potential approximation provides the effective potential for a given classical potential, as stressed by Shepard *et al* [17]. Furthermore, they propose an interesting connection between the Wegner-Houghton sharp-cutoff ERG equation for the local potential [4, 15] and an integral equation for the effective potential. Essentially, they "bootstrap" the relation between the classical and the one-loop effective potential to define a self-consistent "Dyson-Schwinger effective potential". This potential constitutes an approximate solution of the exact renormalization group equation and gives rise to equations that can be related to the Dyson-Schwinger equations. We believe that Shepard *et al*'s interpretation of the exact renormalization group as a non-perturbative renormalization of coupling constants is very interesting.

However, Shepard *et al* [17] restrict their study to the $\phi^4$-theory and to two equations, for the mass and coupling-constant renormalization. These equations are closed, but only because Shepard *et al* arbitrarily change the actual equations derived from the integral equation for the effective potential. This potential actually involves an infinite number of coupling constants, which need to be determined with an infinite number of equations. We elaborate on Shepard *et al*'s idea and we are led to improve the integral equation itself. Our self-consistent effective potential, which involves an infinite number of coupling constants, satisfies a second-order differential equation that provides a new non-perturbative method of renormalization. We present here the general idea and we construct, as an example, the non-perturbative renormalization of scalar field theory in three dimensions.

Shepard *et al* [17] also carry out numerical calculations, namely, Monte Carlo calculations in a lattice and continuous and lattice ERG calculations (all for the $\phi^4$-theory). Tsypin [18] had calculated before the effective potential of the three-dimensional $\phi^4$-theory with the lattice Monte Carlo method. Various later lattice calculations are presented by Butera and Pernici [19]. Of course, calculations in a lattice are affected by the limitation on the number of sites

or other issues. In Shepard *et al*'s continuous ERG calculations, a different problem arises: in the decomposition of the range of the field, which is finite, there appear instabilities at the end of the range. In connection with this problem, we show that a proper treatment of the boundary conditions for large fields is mandatory in our approach.

The standard calculation of the *fixed point* effective potential with the ERG [4, 15] boils down to the solution of a second order differential equation for the effective potential. This equation indeed requires a careful study of the the boundary conditions for large fields. Morris [15] explains how to obtain the large-field asymptotic form of the fixed-point effective potential and, thus, how to resolve the ambiguity in the boundary conditions for vanishing field. The differential equation for the not fixed-point effective potential that we obtain is different from the fixed-point equation in Refs. [4, 15], but it also lends itself to a successful study of the boundary conditions needed for finding physical solutions.

The effective potential can also be calculated in perturbation theory [10]. We discuss the connection of our non-perturbative calculation in three dimensions with perturbative calculations [20–25], which are still ongoing work [26]. This connection is indeed fruitful and unveils advantages and disadvantages of each approach. In contrast, the connection between the fixed-point ERG effective potential and perturbative renormalization is indirect (see the discussion by Morris [27]).

Our approach begins with an arbitrary classical potential but obtains indirect information about it. We shall discover that a natural solution for our formulation of the Dyson-Schwinger effective potential involves, for a single scalar field $\phi$ in three dimensions, not just the $\phi^4$-theory considered by Shepard *et al* [17] but the full renormalizable $\phi^6$-theory. The $\phi^6$-theory has two *relevant* control parameters and encompasses a broader range of applications, including the description of *tricritical* behavior [16, 28, 29].

The tricritical behavior of three-dimensional scalar field theory features among the multi-critical RG fixed points in dimension $d > 2$, which have been the focus of recent interest, e.g., in the work of Codello *et al* in Refs. [30, 31] and in references therein. These references apply a combination of perturbation theory and non-perturbative methods based on conformal field theory. A possible connection of these methods with our approach to the non-perturbative calculation of the effective potential should be very interesting.

Perturbative calculations in the epsilon expansion for multi-critical RG fixed points are compared by O'Dwyer and Osborn [32] with the ERG equation in the local-potential approximation (and beyond). The connection of the ERG equations with dimensional regularization is further studied by Baldazzi *et al* [33]. These works are possibly related to the comparison made in Sect. 6.1 between our non-perturbative approach and perturbation theory. However, we do not employ the epsilon expansion, which turns out to be problematic.

Non-perturbative methods in field theory, namely, the Dyson-Schwinger method and the ERG method, are gaining an increasingly important role in a variety of problems in quantum field theory and, because of this, computer packages that can work out both types of equations have been built [34]. The computational power of such packages can possibly be used to construct approximations of the effective potential better than ours. This goal is beyond the scope of the present work.

The effective potential in field theory is especially useful to study symmetry breaking. We are here mainly concerned with the symmetric phase and the onset of symmetry breaking, in terms of the massless limit. The solution of the ERG equation is more complicated in broken symmetry phases. Notwithstanding, the ERG can be applied to the study of the effective potential in broken symmetry phases [35, 36]. The symmetry that we have in the theory of a single scalar field is very simple, namely, the $\mathbb{Z}_2$ $\phi$-reflection symmetry. Nevertheless, the phase structure near a tricritical point is quite involved [16, 28, 29]. Theories with several fields, including fermionic fields, and with various discrete or continuous symmetries have many applications

but require an ample study. A detailed study of the various aspects of symmetry breaking is beyond the scope of the present work.

Here is our plan. We introduce in Sect. 2 Shepard *et al*'s effective potential, with a notable improvement. In Sect. 3, the integral equation fulfilled by this potential, in three dimensions, is transformed into an ordinary differential equation for non-perturbative renormalization. The mass renormalization is especially important and is studied in Sect. 3.1. The differential equation for the effective potential is thoroughly studied in Sect. 4, which includes: a natural but special solution in the form of a sixth-degree polynomial (Sect. 4.1), the general solution (Sect. 4.2), and a study in Sect. 5 of its asymptotic behavior, based on the differential equation itself, whence follow some numerical examples (Sect. 5.1). A comparison of numerical results of our approach with numerical results of other authors is made in Sect. 6. This section includes Sect. 6.1, containing a comparison with results of perturbation theory and a short discussion of its scope. Finally, because the massless limit is most problematic in all these approaches, Sect. 6.2 contains some remarks about it. We end with a general discussion (Sect. 7). In the appendix, we derive some useful perturbative renormalization formulas for $(\lambda\phi^4 + g\phi^6)_3$ theory.

## 2 The Dyson-Schwinger effective potential

Let us first introduce the "Dyson-Schwinger effective potential", as defined by Shepard *et al* [17]. Starting from the equation for the regularized one-loop effective potential in terms of the classical potential, namely,

$$U_1(\phi) = \frac{1}{2}\int_0^{\Lambda_0} \frac{d^d k}{(2\pi)^d}\, \ln\left[k^2 + U_{\text{clas}}''(\phi)\right].\tag{1}$$

Shepard *et al* heuristically define the "Dyson-Schwinger effective potential" $U_{\text{DS}}$ as the solution of the integro-differential equation

$$U_{\text{DS}}(\phi) = U_{\text{clas}}(\phi) + \frac{1}{2}\int_0^{\Lambda_0} \frac{d^d k}{(2\pi)^d}\, \ln\left[k^2 + U_{\text{DS}}''(\phi)\right].\tag{2}$$

Shepard *et al* show that the derivatives with respect to $\phi$ of this equation yield equations analogous to the standard Dyson-Schwinger equations. Furthermore, they connect the equation with the sharp-cutoff ERG equation, written as

$$\frac{dU(\phi,\Lambda)}{d\Lambda} = -\frac{A_d}{2}\,\Lambda^{d-1}\,\ln\left[\Lambda^2 + U''(\phi,\Lambda)\right],\tag{3}$$

where $A_d = \int d\Omega_d/(2\pi)^d$ (the angular integral such that $d^d k/(2\pi)^d = A_d\, k^{d-1}$, for any integrand that only depends on $k$). Indeed, Shepard *et al* regard Eq. (2) as a crude integration of the ERG equation (3), "namely one in which the integration of the differential equation ... is carried out in a single step."

Actually, it is possible to exactly integrate Eq. (3), that is to say, it is possible to transform it into an integral equation, namely,

$$U(\phi,\Lambda) = U(\phi,\Lambda_0) + \frac{1}{2}\int_\Lambda^{\Lambda_0} \frac{d^d k}{(2\pi)^d}\, \ln\left[k^2 + U''(\phi,k)\right].\tag{4}$$

It becomes, in the limit $\Lambda \to 0$,

$$U(\phi,0) = U(\phi,\Lambda_0) + \frac{1}{2}\int_0^{\Lambda_0} \frac{d^d k}{(2\pi)^d}\, \ln\left[k^2 + U''(\phi,k)\right].\tag{5}$$

This equation is similar to Eq. (2), provided that we identify $U_{\text{DS}}(\phi) = U(\phi, 0)$ and $U_{\text{clas}}(\phi) = U(\phi, \Lambda_0)$. Of course, the second derivative $U''$ in the integrand now depends on the integration variable, in contrast to $U''_{\text{DS}}$ in Eq. (2). It is the dependence of $U''$ on the integration variable that makes it difficult to solve the integro-differential equation (4) to find $U(\phi, \Lambda)$ and hence $U(\phi, 0)$. This is the effective potential that we seek, but we keep the subscript "DS" to emphasize that by using integral equations we can make connections to the Dyson-Schwinger equations.

The assumption that leads to Eq. (2) is that one can replace $U''(\phi, k)$ in the integral in the right-hand side of Eq. (5) by its value at the lower integration limit. Notice that, if we replace $U''(\phi, k)$ in the integral in Eq. (5) by its value at the upper integration limit, then, with the identification $U_{\text{clas}}(\phi) = U(\phi, \Lambda_0)$, we simply have that $U(\phi, 0)$ is the one-loop effective potential. In this case, the identification $U(\phi, \Lambda_0)$ with $U_{\text{clas}}(\phi)$ is natural as a renormalization procedure: the UV cutoff $\Lambda_0$ has to be absorbed in the parameters present in $U_{\text{clas}}$ for the theory to be renormalizable and the limit $\Lambda_0 \to \infty$ to be possible [10]. The exact solution of Eq. (5) yields a transformation of $U_{\text{clas}}(\phi)$ to $U_{\text{DS}}(\phi)$ that amounts to a non-perturbative renormalization of coupling constants.

As noted by Shepard *et al* [17], the second derivative of Eq. (2) with respect to $\phi$ at $\phi = 0$ can be somehow connected with the "cactus approximation" to the Dyson-Schwinger equation for the two-point function, namely, with the standard "gap equation". This equation is a severe truncation of the Dyson-Schwinger equation, as is well known and concisely shown by Swanson [11, §5.4.1]. Actually, the second derivative of Eq. (2) with respect to $\phi$ at $\phi = 0$ gives

$$m_{\text{DS}}^2 = m_0^2 + \int_0^{\Lambda_0} \frac{d^d k}{(2\pi)^d} \frac{12\lambda_{\text{DS}}}{k^2 + m_{\text{DS}}^2} \,, \tag{6}$$

where $m_0^2 = U''_{\text{clas}}(0)$, $m_{\text{DS}}^2 = U''_{\text{DS}}(0)$, and $\lambda_{\text{DS}} = U''''_{\text{DS}}(0)/4!$. The gap equation results from Eq. (6) when one *substitutes* inside the integral the renormalized coupling $\lambda_{\text{DS}}$ by the bare coupling $\lambda_0 = U''''_{\text{clas}}(0)/4!$ but keeps $m_{\text{DS}}$ intact. If we also substitute $m_{\text{DS}}$ by $m_0$, then it is like putting $U''_{\text{clas}}$ instead of $U''_{\text{DS}}$ in the integral in Eq. (2) before taking the second derivative, and thus we just have the one-loop correction to the bare mass.

The integrated ERG equation (5) yields, by taking its second derivative with respect to $\phi$ at $\phi = 0$:

$$U''_{\text{DS}}(0) = U''_{\text{clas}}(0) + \frac{1}{2} \int_0^{\Lambda_0} \frac{d^d k}{(2\pi)^d} \frac{U''''(0, k)}{k^2 + U''(0, k)} \,, \tag{7}$$

where $U''(0, k)$ and $U''''(0, k)/4!$ represent the scale-dependent square mass and fourth-order coupling constant [naturally, we assume $U'''(0, k) = 0$]. This equation implies that $m_{\text{DS}}^2 > m_0^2$ but hardly gives more information, unless we are able to say something about the dependence on $k$ of the integrand in the right-hand side.

To obtain the gap equation from Eq. (7), we have to take $U''(0, k) = U''(0, 0) = m_{\text{DS}}^2$ but $U''''(0, k) = U''''(0, \Lambda_0) = 4! \lambda_0$. We can understand this substitution as an instance of a sort of hybrid approach to Eq. (5), in which one replaces $U''(\phi, k)$ in the integrand by its value neither at the upper integration limit nor at the lower limit but by an expression that combines both values, namely, which combines the renormalized mass, which belongs to the lower limit, with the bare coupling, which belongs to the upper limit. This idea motivates us to find a systematic approximation to Eq. (5) that considers the change of $U''(\phi, k)$ when $k$ is brought from $\Lambda_0$ down to 0.

The assumption that one can replace $U''(\phi, k)$ in the integral in the right-hand side of Eq. (5) by its value at the lower integration limit is consistent with Shepard *et al*'s idea [17] of integrating the ERG equation (3) "in a single step." However, the assumed constancy of $U''(\phi, k)$ in the integration over $k$ produces certain error. Naturally, if we manage somehow

to account for the dependence of $U''(\phi, k)$ on $k$, at least partially, we must have a better approximation. A simple way of doing it consists in a two-step integration, in which we split the integration range $[0, \Lambda_0]$ in two parts, using $U_{\mathrm{clas}}$ in the upper part and $U_{\mathrm{DS}}$ in the lower part. This procedure still gives an equation of the type of Eq. (2), that is, an equation with only two functions, namely, the unknown function $U_{\mathrm{DS}}$ and the input function $U_{\mathrm{clas}}$. Of course, a multi-step integration would be more accurate but would introduce other functions and thus it would be considerably more complicated; unless the procedure were carried out in a fully numerical fashion.

A multi-step integration of Eq. (5) is a numerical integration of the partial differential equation (3) by discretization of $\Lambda$ (and possibly $\phi$). Such numerical integration can be performed in several ways [17, 36, 39, 40]. However, it inevitably demands an assumption about the large-field behavior. Actually, the range of the field is often truncated to make it finite and the corresponding boundary condition is somewhat arbitrary. Therefore, it seems that a rather extensive study of the relation between the various boundary conditions and of the influence of changes in the range of the field should be carried out to assess the validity of the procedure. Unfortunately, numerical methods deprive us of intuition and, hence, require extra work, which can be avoided with the insight gained from analytic methods. Our aim is to provide this insight.

Naturally, a multi-step integration allows better control of numerical error, which is somewhat undefined in our approach. However, the error is usually estimated by comparison, even in completely numerical approaches. The absence of error estimates in our approach is compensated by the convenient formulation in terms of integral equations related to the Dyson-Schwinger equations and the later connection with perturbative renormalization.

Both Eq. (2) or the improved version with a two-step integration allow us to express the renormalized couplings in $U_{\mathrm{DS}}$ in terms of the bare couplings in $U_{\mathrm{clas}}$. Shepard *et al* relate Eq. (2) to the Dyson-Schwinger equations but the improved version is a better approximation to these equations. Indeed, the Dyson-Schwinger equation for the two-point function, for example, actually contains the scale-dependent four-point vertex [11, §5.4.1], like Eq. (7). That scale dependence is partially included in the improved version, which can be understood as a better approximation to the corresponding Dyson-Schwinger equation.

Moreover, Eq. (2) leads to Eq. (6), which cannot be valid in the massless limit $m_{\mathrm{DS}} \to 0$ (in dimension $2 < d < 4$) because, in this limit, the renormalized coupling constant also vanishes [37, 38]. Because of Eq. (6), the vanishing of $\lambda_{\mathrm{DS}}$ implies that $m_0 = 0$, contradicting that $m_0$ is an arbitrary parameter. In fact, we must choose a value $m_0^2 < 0$ to have the massless renormalized theory, as deduced from general principles [37, 38] and from Eq. (7). This question is further discussed in Sect. 4.1, in regard to mass renormalization in three dimensions.

To obtain a concrete relation between $U_{\mathrm{DS}}$ and $U_{\mathrm{clas}}$, we shall confine ourselves to a given dimension $d$. Our main interest in this paper is the case $d = 3$. The integrals in Eq. (2) or in our improved version are easily carried out in (integer) dimension $d$, like in the calculation of the one-loop effective potential $U_1$ with Eq. (1). The result of such integration will just be a nonlinear second-order differential equation for $U_{\mathrm{DS}}$, given $U_{\mathrm{clas}}$.

## 3 Equation for the Dyson-Schwinger effective potential in 3d

In $d = 3$, when we integrate over $k$ in Shepard *et al*'s equation (2), we obtain:

$$U_{\mathrm{DS}}(\phi) = U_{\mathrm{clas}}(\phi) + \frac{1}{4\pi^2} \Lambda_0 U''_{\mathrm{DS}}(\phi) - \frac{1}{12\pi} [U''_{\mathrm{DS}}(\phi)]^{3/2}. \tag{8}$$

In this equation we have suppressed the part that only depends on $\Lambda_0$, which is divergent when $\Lambda_0 \to \infty$, and we have neglected corrections that are suppressed by inverse powers of

$\Lambda_0$. The term proportional to $\Lambda_0$ must be kept, but its coefficient depends on the formulation of the ERG, which is given by Eqs. (3) and (4) (a different formulation can be employed, such as, for example, the lattice regularization method of Shepard *et al* [17]). This coefficient dependence is part of the scheme dependence of the ERG [8, 9], which is here condensed in just one parameter. Naturally, it is required that the potential be convex, namely, $U''_{\mathrm{DS}}(\phi) \geq 0$, to have a well defined 3/2-power in Eq. (8). This might seem to make Eq. (8) inapplicable to broken symmetry phases. However, the ERG equation (3) can actually be proved to have convex solutions even in broken symmetry phases [35, 36].

Now we intend to improve on Eq. (2) by means of a two-step integration in Eq. (5), namely, by splitting the integration range $[0, \Lambda_0]$ into an upper part, where we replace $U(\phi, k)$ by $U_{\mathrm{clas}}(\phi)$, and a lower part, where we replace $U(\phi, k)$ by $U_{\mathrm{DS}}(\phi)$. Let the dividing point be $\Lambda_i$. A short calculation gives (in $d = 3$), instead of Eq. (8),

$$U_{\mathrm{DS}}(\phi) = U_{\mathrm{clas}}(\phi) + \frac{1}{4\pi^2}(\Lambda_0 - \Lambda_i)U''_{\mathrm{clas}}(\phi) + \frac{1}{4\pi^2}\Lambda_i U''_{\mathrm{DS}}(\phi) - \frac{1}{12\pi}[U''_{\mathrm{DS}}(\phi)]^{3/2}. \quad (9)$$

The value of $\Lambda_i$ is at our disposal. However, the ERG parameter actually is $t = \log(\Lambda_0/\Lambda)$ and most of the coupling constant "running" takes place for large $t$, that is to say, for $\Lambda \ll \Lambda_0$. Thus, we choose $\Lambda_i \ll \Lambda_0$. Therefore, it seems reasonable to remove small terms from Eq. (9) and write

$$U_{\mathrm{DS}}(\phi) = U_{\mathrm{clas}}(\phi) + \frac{1}{4\pi^2}\Lambda_0 U''_{\mathrm{clas}}(\phi) - \frac{1}{12\pi}[U''_{\mathrm{DS}}(\phi)]^{3/2}, \quad (10)$$

which is only slightly different from Eq. (8) but is considerably more accurate, as we shall see.

Equation (10) can be suitably transformed by rescaling the field and the potential by powers of $\Lambda_0$, that is to say, by taking $\Lambda_0$ as the scale of reference and making $\Lambda_0 = 1$. Indeed, defining

$$x = \phi/\Lambda_0^{1/2}, \quad \tilde{U}(x) = U(\phi)/\Lambda_0^3,$$

equation (10) adopts the form:

$$U_{\mathrm{DS}}(x) = U_{\mathrm{clas}}(x) + \alpha U''_{\mathrm{clas}}(x) - \frac{1}{12\pi}[U''_{\mathrm{DS}}(x)]^{3/2}, \quad (11)$$

where $\alpha = 1/(4\pi^2)$ and we have suppressed the tildes for simplicity. Equation (11) is the basis of our approach.

Both Eqs. (8) and (10) are self-consistent. In the former, the self-consistency is due to the substitution of $U_{\mathrm{clas}}$ by $U_{\mathrm{DS}}$ in the integral for the one-loop effective potential (1) to have the original equation (2). If $U_{\mathrm{clas}}$ is characterized by some coupling constants and we assume that $U_{\mathrm{DS}}$ can be written in terms of the same coupling constants, the substitution of $U_{\mathrm{clas}}$ by $U_{\mathrm{DS}}$ amounts to the substitution of the bare coupling constants by the renormalized ones. Such substitution is actually made in perturbation theory through the definition of the *renormalized loop expansion*, as explained by Zinn-Justin in Ref. [38, p. 246 ff]. In this case, the effect is that only *superficially divergent* Feynman diagrams need to be renormalized, because the divergent subdiagrams are taken into account self-consistently. Thus, Eq. (8) is connected with perturbation theory.

In contrast, Eq. (10) is not connected with perturbation theory. Next, we study various forms of relationship between bare and renormalized parameters, in the simple case of mass renormalization.

## 3.1 Perturbative and non-perturbative mass renormalization

We have three different equations for the effective or renormalized potential, namely, the perturbative one-loop equation, Shepard *et al*'s Eq. (8), and our improved Eq. (10). We can

compare them in the simple case of mass renormalization, whose equation is obtained by taking the second derivative of the effective potential. With the definitions $U''_{\text{clas}}(0) = m_0^2$, $U''''_{\text{clas}}(0) = 4! \lambda_0$, $U''_{\text{eff}}(0) = m^2$, and $U''''_{\text{eff}}(0) = 4! \lambda$, we have first the perturbative one-loop equation:

$$m^2 = m_0^2 + \frac{6\Lambda_0}{\pi^2} \lambda_0 - \frac{3}{\pi} m_0 \lambda_0 \,. \tag{12}$$

Shepard *et al*'s Eq. (8) gives instead:

$$m^2 = m_0^2 + \frac{6\Lambda_0}{\pi^2} \lambda - \frac{3}{\pi} m \lambda \,. \tag{13}$$

Eq. (10) gives:

$$m^2 = m_0^2 + \frac{6\Lambda_0}{\pi^2} \lambda_0 - \frac{3}{\pi} m \lambda \,. \tag{14}$$

In addition, we have the gap equation, namely,

$$m^2 = m_0^2 + \frac{6\Lambda_0}{\pi^2} \lambda_0 - \frac{3}{\pi} m \lambda_0 \,. \tag{15}$$

All these four mass renormalization equations look similar but are quite different.

Shepard *et al*'s mass renormalization equation (13) coincides with the renormalized one-loop equation (perturbation theory is further discussed in Sect. 6.1). This equation is wrong in the massless $m \to 0$ limit, because in this limit $\lambda \to 0$ as well, as will be shown shortly. The gap equation involves the renormalized mass but the bare quartic coupling. Nevertheless, the gap equation is reasonable in the massless limit: it implies a relation between the bare mass and quartic coupling, namely,

$$m_0^2 = -\frac{6\Lambda_0}{\pi^2} \lambda_0 \,, \tag{16}$$

which is fine, as shown below. This same relation follows from Eq. (14) in the massless limit.

Let us consider the linearized Wegner-Houghton ERG directly in parameter space. The nonlinear equations including up to the sextic coupling are written by Haagensen *et al* [6, §4]. Linearizing these equations, we obtain that the sextic coupling is constant, hence null, and we have just two equations:

$$\frac{d\sigma}{dt} = 2\sigma + u \,, \tag{17}$$

$$\frac{du}{dt} = u \,, \tag{18}$$

where $t = \ln(\Lambda_0/\Lambda)$, $\sigma = m^2/\Lambda^2$, and $u = 6\lambda/(\pi^2\Lambda)$. System (17,18) has eigenvalues $\{2, 1\}$ and corresponding eigenvectors $\{(1, 0), (-1, 1)\}$ (in this regard, see Ref. [28, §5.3]). Therefore, the system is solved by defining the new variable $v = \sigma + u$, giving

$$\frac{dv}{du} = 2\frac{v}{u} \,.$$

Its solution is

$$v = \sigma + u = K u^2 \,,$$

with an arbitrary constant $K \geq 0$.

Geometrically, these RG trajectories are parabolas in the $(\sigma, u)$ plane, which have horizontal axes and are tangent to the line $\sigma = -u$ at the origin. This RG flow is only valid in the neighborhood of the origin (small $m_0$ and $\lambda_0$) and for small $t$, because both $\sigma$ and $u$ are driven

away from the origin for growing $t$. In contrast, the corresponding values of $m^2$ and $\lambda$ go to finite limits when $t \to \infty$ ($\Lambda \to 0$), namely,

$$m^2 = m_0^2 + \frac{6\Lambda_0}{\pi^2}\lambda_0 \,, \tag{19}$$

and $\lambda = \lambda_0$ (actually, $\lambda$ is constant for any $\Lambda$). We find that Eq. (16) is indeed required to have $m = 0$ in Eq. (19). Naturally, Eq. (19) is not exact (and $\lambda \neq \lambda_0$), because the linearized ERG is not applicable for $t \gg 1$ ($\Lambda \ll \Lambda_0$).

Nevertheless, the full nonlinear ERG gives a massless limit that may not depart much from Eq. (16). For example, Hasenfratz and Hasenfratz [4, §5.1] choose $\lambda_0 = 3\Lambda_0/4$ (in our notation) and compute the value of $m_0^2$ that makes $m = 0$, by solving the nonlinear ERG equation and tuning $m_0^2$ to hit the critical surface. They find $m_0^2/\Lambda_0^2 = -0.4576$, whereas Eq. (16) gives $m_0^2/\Lambda_0^2 = -9/(2\pi^2) = -0.4559$.

The gap equation (15) is a better mass renormalization equation than Eq. (19), and it reduces to Eq. (19) when $m \ll \Lambda_0$. Our equation (14) reduces to the gap equation when $\lambda = \lambda_0$ (which is correct under the linearized ERG). Thus, those three mass renormalization equations can be considered as successively better approximations to the exact mass renormalization equation. Our equation (10) actually implies that $\lambda \to 0$ in the massless limit (Sect. 4), as expected. Indeed, $\lambda$ must vanish at the non-trivial fixed point of the ERG, namely, at the Wilson-Fisher fixed point (which does not appear in the linearized form, of course, but is easily found with the non-linear ERG [1, 6]). Given that $u = 6\lambda/(\pi^2\Lambda)$ stays finite at the non-trivial fixed point, $\lambda$ must vanish with $\Lambda$.

In conclusion, both the gap equation (15) and our equation (14) are fine in the massless limit whereas Eq. (13) is not. Consequently, our general equation (10) for the full effective potential definitely improves on Shepard *et al*'s Eq. (8).

## 4 Solution of the differential equation

Our approximation to the Dyson-Schwinger effective potential is given by the nonlinear second-order differential equation (11), which contains the input function $U_{\text{clas}}$. This second-order differential equation can be rewritten in several ways. One obvious way consists in solving for $U_{\text{DS}}''$. Another is to redefine the dependent variable as

$$w = U_{\text{DS}} - U_{\text{clas}} - \alpha U_{\text{clas}}'' \,,$$

to have the second derivative of the dependent variable expressed as the sum of two terms, one with the dependent variable and another with the independent variable, namely,

$$w''(x) = [-12\pi w(x)]^{2/3} - U_{\text{clas}}''(x) - \alpha U_{\text{clas}}''''(x). \tag{20}$$

Either way, we can think of an analogy with the problem of the one-dimensional motion of a particle under an arbitrary force in mechanics, supposing that the dependent variable represents the position and the independent variable represents the time. This analogy is more useful in the form (20), which corresponds to the sum of a conservative force and a time-dependent force in mechanics. In fact, we can add $-(U_{\text{clas}}'' + \alpha U_{\text{clas}}'''')(0)$ to the conservative force and subtract it from the "time-dependent force", so that the latter initially vanishes.

We have two integration constants that can be determined by standard "initial" conditions at $x = 0$, namely, by the values of the dependent variable and its first derivative at $x = 0$. Naturally, the presence of the "time-dependent force" in Eq. (20) implies that there is no "energy" first integral; unless there is no such force, as happens when $U_{\text{clas}}(x)$ is a quadratic polynomial, that is to say, in the trivial Gaussian case. The first derivative at $x = 0$ is imposed by

the reflection symmetry of the potential, namely, $U'_{\mathrm{DS}}(0) = 0$ or $w'(0) = 0$ (we assume that $U_{\mathrm{clas}}(x)$ is symmetric, of course).

Differential equation (11) can also be transformed into a simpler second order equation by taking its derivative with respect to $x$ and replacing $U'_{\mathrm{DS}}$ by a new dependent variable (as done to the differential equation for the ERG fixed-point local potential [4]). With this procedure, the initial conditions consist of the values $U'_{\mathrm{DS}}(0)$ and $U''_{\mathrm{DS}}(0)$. As said above, the former must vanish whereas the latter has an important physical meaning: it gives the field-theory mass (or correlation length).

A general problem with nonlinear differential equations such as (11) is that they can exhibit singularities which depend on the initial conditions and are called "spontaneous" or "movable" singularities. Note that such singularities can occur in one equation for most initial conditions. This is the case of the differential equation for the ERG fixed-point local potential [4, 15], and the presence of singularities for most initial conditions actually allowed Morris [15] to *determine* the initial conditions, by imposing the absence of singularities. Equation (20) can develop a sort of singularities that depend on the initial conditions, although the absence of them does not fully determine the initial value $w(0)$.

Indeed, the solution of Eq. (20) may be *non extendable* whenever $w$ vanishes, because of the 2/3-power. In general, we must choose the initial condition $w(0) < 0$, so that the 2/3-power is well defined. This condition is equivalent to the convexity of $U_{\mathrm{DS}}$. For some initial condition $w(0) < 0$, the dynamics given by Eq. (20) is intuitive: the "conservative force" drives $w$ towards zero whereas the "time-dependent force" drives $w$ towards more negative values, because we assume that the even derivatives of $U_{\mathrm{clas}}$ are positive at $x = 0$. The "time-dependent force" initially vanishes, whereas the first "force" is initially larger the larger is $-w(0)$ and makes $w$ go towards zero for an interval of $x$. If $w$ does hit zero with non-vanishing "velocity" $w'$, then the solution stops at the corresponding value of $x$.

Nevertheless, we can have non-singular and extendable solutions, provided that $-w(0)$ is small and non-vanishing (Sect. 4.2). At any rate, our intention is not to carry out a detailed study of the singularities of the general solution of Eq. (11) and we proceed on a different tack. First, we present the solution for a particular form of $U_{\mathrm{clas}}$ (Sect. 4.1). Second, we consider in Sect. 4.2 the local properties of the general solution at $x = 0$, taking into account what is learnt from the particular solution. In Sect. 5, we combine this study with the study of asymptotic properties.

## 4.1 Polynomial solution

In a quantum field theory in some dimension $d$, the input function $U_{\mathrm{clas}}(x)$ is a polynomial and the renormalizability of the effective potential implies a bound to the degree of the polynomial that depends on $d$ [10, 37, 38, 42]. In $d = 3$, the maximum degree is 6. If we set $\alpha = 0$ and put $U_{\mathrm{clas}}(x) \propto x^6$ in Eq. (11), then we have a solution $U_{\mathrm{DS}}(x) \propto x^6$, because the consequent homogeneity in the variable $x$ allows us to adjust the coefficients to satisfy Eq. (11). All of this suggests us to choose $U_{\mathrm{clas}}(x)$ as a sixth-degree polynomial, with only even powers, to make it symmetrical; namely,

$$U_{\mathrm{clas}}(x) = g_0 \, x^6 + \lambda_0 \, x^4 + r_0 \, x^2 + b_0 \,. \tag{21}$$

Hence, we seek a particular solution of Eq. (11) in the form of a symmetrical sixth-degree polynomial, namely,

$$U_{\mathrm{DS}}(x) = g \, x^6 + \lambda \, x^4 + r \, x^2 + b \,. \tag{22}$$

Some simple but lengthy algebra shows that this is indeed a solution, provided that a set of 7 algebraic equations for the 8 coefficients is satisfied. The 7 algebraic equations are easily obtained by solving for $[U''_{\mathrm{DS}}(x)]^{3/2}$ in Eq. (11), squaring, and equating the coefficients of the

two twelfth-degree polynomials in $x$. By squaring, we introduce sign ambiguities, but the signs can be determined by always referring to the positive square root in Eq. (11). Let us discuss the nature of the equations.

First of all, the solution could actually be determined, because in the set of 7 algebraic equations only appears the combination $b - b_0$, and therefore the number of independent unknowns is 7 as well. Naturally, we set $b_0 = 0$ (this additive constant in $U_{\text{clas}}$ is not significant). In general, $b \neq 0$. Let us consider the equations given by each subsequent power of $x$.

- The first and simplest equation ($x^0$) can be written as

$$b = 2\alpha r_0 - \frac{2^{1/2}}{6\pi} r^{3/2}, \tag{23}$$

  which just expresses the unimportant constant $b$ in terms of $r_0$ and $r$.

- Next equation ($x^2$), after substituting for $b$ according to Eq. (23), can be written as

$$r - r_0 = 12\alpha\lambda_0 - \frac{3}{2^{1/2}\pi} r^{1/2}\lambda. \tag{24}$$

  This equation is just another form of writing the mass renormalization equation (14), which is a general consequence of Eq. (10) and is in accord with the linearized ERG, as explained in Sect. 3.1.

- Next equation ($x^4$), after the pertinent substitutions, gives $\lambda - \lambda_0$ in terms of $r, \lambda, g$ (we assume that $r = U_{\text{DS}}''(0)/2 \neq 0$):

$$\lambda - \lambda_0 = 30\alpha g_0 - \frac{3r^{1/2}}{2\,2^{1/2}\pi}(5g + 3r^{-1}\lambda^2). \tag{25}$$

  It is related to one-loop $\lambda$-renormalization equations, namely, to Eqs. (A.5) or (A.9) to $O(\hbar)$, but it matches neither.

- Next equation ($x^6$) gives $g - g_0$ in terms of $r, \lambda, g$:

$$g - g_0 = \frac{9}{2\,2^{1/2}\pi}\lambda\, r^{-1/2}(-5g + r^{-1}\lambda^2). \tag{26}$$

  It matches a $g$-renormalization equation, namely, Eq. (A.10) to $O(\hbar)$. Notice that it does not involve $\alpha$, which is associated with terms divergent in the limit $\Lambda_0 \to \infty$.

The last three equations allow us to express the bare parameters as explicit functions of the renormalized parameters. Solving for the renormalized parameters $r, \lambda, g$, we could express them in terms of the bare ones. Interestingly, the next two equations ($x^8, x^{10}$) reduce to one equation, upon replacing $r_0, \lambda_0, g_0$ in accord to the preceding equations. The equation is:

$$5rg = 3\lambda^2. \tag{27}$$

The remaining seventh equation ($x^{12}$) can be written as

$$(30g)^{3/2} = -12\pi(g - g_0), \tag{28}$$

or

$$g_0 = g + \frac{30^{3/2}}{12\pi} g^{3/2}. \tag{29}$$

This relation is precisely the one that is obtained by making $\alpha = 0$ and setting $U_{\text{clas}}(x) = g_0 x^6$ and $U_{\text{DS}}(x) = g\, x^6$ in Eq. (11).

Excluding $b$ (which we can calculate at the end), we have 5 independent equations for $r, \lambda, g$ and $r_0, \lambda_0, g_0$. Therefore, we can choose one variable, say $r$, and solve for the others in terms of it. The simplest procedure seems to be the following. We can eliminate $g_0$ between Eqs. (26) and (28), so that we have an equation for $r, \lambda, g$. This equation turns out to be satisfied provided that Eq. (27) is satisfied. Therefore, we can choose arbitrarily two variables, say $r$ and $\lambda$, and then $g$ is given by Eq. (27). As regards the application to the theory of phase transitions, potential (22) can have three minima, that is to say, three phases, and two relevant variables are necessary for tricritical behavior [28,29] (also Ref. [38, p. 608]).

It seems more natural to let the input parameters be the bare ones. However, the constraint that affects $r, \lambda, g$ implies a constraint on $r_0, \lambda_0, g_0$, leaving independent only two of them, which can be $r_0$ and $\lambda_0$. (However, we shall set $r$ and $\lambda_0$ in Sect. 6, to compare with the calculations of Shepard *et al* [17].) It may seem odd that $g_0$ is determined and non-vanishing, because it vanishes in the $\phi^4$-theory. We can legitimately ask what happens if we arbitrarily fix the three bare parameters $r_0, \lambda_0, g_0$ and just solve for the renormalized parameters by means of Eqs. (24), (25), and (26). This possibility is discussed below.

Let us now study the meaning of Eq. (27). When this equation is fulfilled, $U''_{\text{DS}}(x)$ is a perfect square, namely,

$$U''_{\text{DS}}(x) = 2(r + 3\lambda x^2)^2/r\,. \tag{30}$$

Naturally, then we have that

$$[U''_{\text{DS}}(x)]^{3/2} = (2/r)^{3/2}\,|r + 3\lambda x^2|^3\,,$$

and it is a polynomial in $x$, provided that $r + 3\lambda x^2 \geq 0$, as occurs when $r, \lambda \geq 0$. Note that the potential can be convex with $\lambda < 0$, provided that $5rg \geq 3\lambda^2$, and then $U''_{\text{DS}}(x)$ in Eq. (30) vanishes for some value of $x$. Nevertheless, we consider here that $\lambda > 0$.

We conclude that Eq. (27) or, rather, its expression in terms of the bare coupling constants is the closure equation for polynomial solutions of Eq. (11) with input (21). If Eq. (27) holds, then Eq. (11) boils down to the set of equations from (23) to (26) [with the substitution $g = 3\lambda^2/(5r)$]. The role of this set of equations is further clarified by the study of the general solution of Eq. (11).

## 4.2 General solution

So far, we have found an interesting but particular solution of Eq. (11), namely, a sixth-degree polynomial with coefficients that can be determined sequentially. The procedure employed suggests us to look for the general solution of Eq. (11) as a power series expansion at $x = 0$; namely, we assume that

$$U_{\text{DS}}(x) = \sum_{k=0}^{\infty} c_k x^{2k} \tag{31}$$

and try to determine the coefficients $c_k$ from Eq. (11). The sixth-degree polynomial solution in Eq. (22) corresponds to the truncation to $k = 3$, and we preserve the former names of $c_0, \ldots, c_3$. It is the only *exact* polynomial solution. Naturally, the analyticity assumption implicit in Eq. (31) is only valid if $U''_{\text{DS}}(0) \neq 0$, that is to say, it is not valid at the singular point of Eq. (11). We must caution that the expansion of the effective potential in powers of $x$ ceases to be valid for $x \gg (r/\lambda)^{1/2}$, on general grounds [23,24].

The sixth-degree polynomial solution in Eq. (22) imposes that $U_{\text{clas}}(x)$ is also a sixth-degree even polynomial, with three coefficients and a constraint on them. However, if $U_{\text{clas}}(x)$ is an arbitrary sixth-degree (even) polynomial, then we can still find a solution of Eq. (11), as the *infinite* series in Eq. (31). We try to determine the coefficients $c_k$ sequentially, but we no longer

assume Eq. (27), which makes $c_k$ vanish for $k \geq 4$. Equations (23), (24), and (25) are not modified, but an additional term, with $c_4$, is generated in Eq. (26):

$$g - g_0 = -\frac{45g\lambda}{2\sqrt{2}\pi r^{1/2}} + \frac{9\lambda^3}{2\sqrt{2}\pi r^{3/2}} - \frac{14c_4 r^{1/2}}{\sqrt{2}\pi}. \tag{32}$$

Equations (24), (25), and (32) allow us to express the renormalized parameters $r, \lambda, g$ in terms of the bare ones and $c_4$. To determine $c_4$ we have to look to the following equations. Next one is

$$c_4 = -\frac{3\left(60c_5 r^3 + 112c_4 r^2 \lambda + 75g^2 r^2 - 90gr\lambda^2 + 27\lambda^4\right)}{8\sqrt{2}\pi r^{5/2}}. \tag{33}$$

Naturally, we face the usual problem of non-perturbative approaches: the issue of an infinite tower of equations that has to be truncated somehow to obtain definite solutions. If we just make $c_5 = 0$, then Eqs. (24), (25), (32), and (33) are a closed system for $r$, $\lambda$, $g$, and $c_4$.

Of course, if we make $c_4 = 0$, then Eq. (33) and all the following equations are automatically satisfied provided that Eq. (27) holds, which implies that $c_k = 0$ for $k \geq 4$. With this particular truncation, we have the *exact* sixth-degree polynomial solution. If we make $c_5 = 0$ but assume that $c_4 \neq 0$, then we can solve for $c_4$ in Eq. (33), obtaining:

$$c_4 = -\frac{9\left(5gr - 3\lambda^2\right)^2}{8r^2\left(\sqrt{2}\pi\sqrt{r} + 42\lambda\right)}. \tag{34}$$

Hence, we can substitute for $c_4$ in Eq. (32) and obtain a correction to Eq. (26) that has larger magnitude the larger is the deviation from Eq. (27). By assuming that $c_4 = 0$ in Eq. (32), we are introducing a truncation error, which can be evaluated by computing what $c_4$ should be according to Eq. (34) for the found solution. We can also substitute for $c_4$ in Eq. (32) according to Eq. (34) before solving it. However, if $c_4 \neq 0$, then we do not have that $c_k$ for all $k \geq 5$, and some high-degree coefficients could be large.

For small $|x|$ and fixed $r_0, \lambda_0, g_0$, the recursive reduction of the sequence of $c_k$ to one definite unknown is straightforward, as we now describe. From the two initial conditions for Eq. (11), the non-trivial one is the value of $r$ ($r \neq 0$). Once we have chosen it, we can solve Eq. (24) for $\lambda(r)$, substitute in Eq. (25) and solve for $g(r)$, substitute in Eq. (32) and solve for $c_4(r)$, etc. However, the found solution may not have the right behavior for large $|x|$. Indeed, we have seen at the beginning of Sect. 4 that a singularity appears when the value of

$$-w(0) = \alpha U''_{\text{clas}}(0) - U_{\text{DS}}(0) = 2\alpha r_0 - b = \frac{2^{1/2}}{6\pi}r^{3/2}$$

is large enough. Therefore, the range of $r$ is restricted. The study of the large-$|x|$ behavior leads to further restrictions and, in fact, to a definite solution, as explained in Sect. 5.

Naturally, the variable $r$ plays a special role in the equations for $c_k$, as we know from its physical meaning and also because it features in the denominators of Eqs. (25), (32), and (33). There are two interesting limits, namely, $r \to \infty$ and $r \to 0$. The former is a sort of "classical" limit, that is to say, the limit in which fluctuations are negligible. To understand this limit, it is useful to recall that our coupling constants are dimensionless, after the rescaling by powers of $\Lambda_0$, and the limit $r \to \infty$ is achieved when the dimensional coefficient of the $\phi^2$-term is $\gg \Lambda_0^2$. In principle, if we make $\Lambda_0 \to 0$, then $U_{\text{DS}}(\phi) = U(\phi, 0)$ will approach $U_{\text{clas}}(\phi) = U(\phi, \Lambda_0)$, and the renormalized parameters should tend to the bare ones. However, we have neglected corrections that are suppressed by inverse powers of $\Lambda_0$ in Eq. (8), making that limit non-trivial. The opposite limit $r \to 0$ determines critical behavior, which is further studied in Sect. 6.2.

# 5 Asymptotic behavior for large fields

An interesting question is how the effective potential behaves for large values of the field. In other words, we ask the behavior of the differential equation in the neighborhood of $x = \infty$ or, more rigorously, in the neighborhood of $y = 0$, where $y = 1/x$. Effecting this change of the independent variable, accompanied by a convenient change of the dependent variable, we obtain the associated equation:

$$y^2 g''(y) - 10y g'(y) + 30g(y) = (-12\pi [g(y) - \tilde{u}(y)])^{2/3}, \tag{35}$$

where

$$g(y) = y^6 U_{\text{DS}}(1/y)$$

and

$$\tilde{u}(y) = y^6 \left[ U_{\text{clas}}(1/y) + \alpha \, U''_{\text{clas}}(1/y) \right].$$

Differential equation (35) contains $g'$ in addition to $g''$ and is, therefore, a bit more complicated than the equations in terms of $x$. However, it has some interesting properties at $y = 0$, which we now study.

Naturally, $y = 0$ is a singular point of Eq. (35). Nevertheless, if $\tilde{u}(0)$ is finite, then we can have an analytic solution for $g(y)$, thus justifying the definitions of $g$ and $\tilde{u}$. Of course, $\tilde{u}(0)$ is finite only if the growth of $U_{\text{clas}}(x)$ is $O(x^6)$ at the most, that is to say, only if it has the sextic form (21). We can now expand $g(y)$ in powers of $y$ at $y = 0$, substitute for it in Eq. (35), and solve for the coefficients in sequence, namely, for the sequence of derivatives of $g(y)$ at $y = 0$. Given that the polynomial $\tilde{u}(y)$ has no odd-degree powers, the odd-order derivatives of $g(y)$ vanish (as expected). The even-degree coefficients are expressed in terms of $r_0, \lambda_0, g_0$. In particular, the first coefficient, $g(0)$, is given by the $g$ in Eq. (28).

The power expansion of $g(y)$ at $y = 0$ does not end at any finite power, for generic values of $r_0, \lambda_0, g_0$, unlike in the case that these values satisfy the constraint that makes $U_{\text{DS}}(x)$ a sextic polynomial and, therefore, that it also makes $g(y)$ a sextic polynomial (the constraint appears in Sect. 4.1). Of course, the truncation of the power expansion of $g(y)$ to $O(y^6)$ makes $U_{\text{DS}}(0)$ finite, whereas higher-order terms make it divergent. Naturally, the power expansion of $g(y)$ at $y = 0$ does not have an infinite radius of convergence and does not serve to calculate $U_{\text{DS}}(0)$. Therefore, we cannot *directly* deduce the missing boundary condition at $x = 0$ for Eq. (11). To do it, one possibility is to take $g(y)$ at some value $y$ within the radius of convergence and numerically integrate Eq. (11) from $x = 1/y$ to $x = 0$, with initial conditions $U_{\text{DS}}(x) = x^6 g(1/x)$ and its derivative. Another possibility, which we find more precise, is a sort of asymptotic matching, which is best explained with some examples below (this technique is related to the one employed by Borchardt and Knorr [41] in a related problem, namely, the decomposition of the range of $x$ into $[0, x_0]$ and $[x_0, \infty]$).

It is to be remarked that the phase-space point with $g'(0) = 0$ and $g(0)$ given by Eq. (28) is a singular point of Eq. (35) and there is no unique solution through it. Indeed, the calculation of the power expansion of $g(y)$ ignores possible non-analytic terms. Numerical experiments show that the integration of Eq. (28) with the given boundary conditions is unstable, resulting in oscillations of increasing amplitude. Therefore, the analytic solution is unstable under small perturbations that bring in the non-analytic terms. At any rate, the analytic solution is dependable, because the power series seems to converge well for reasonably large values of $y$ (small $x$).

## 5.1 Asymptotic matching: numerical examples

Let us see how the asymptotic matching works in an example. Hasenfratz and Hasenfratz [4, §5.1] computed the ERG evolution of the case $\lambda_0 = 3/4$ and $g_0 = 10/3$ (in our notation)

along a range of $t = \ln(\Lambda_0/\Lambda)$, with $r$ close to the critical value. These parameter values are a possible choice for us. Unfortunately, Hasenfratz and Hasenfratz did not specify the boundary condition for the potential at $x = \infty$ (the field is, according to their graphs, restricted to the range $x < 0.4$). Moreover, we are interested in the limit $\Lambda \to 0$, and thus it is difficult to compare with their results. Hence, to be able to connect in Sect. 6 with Shepard *et al*'s numerical work [17], we choose

$$U_{\text{clas}}(x) = 0.1x^6 + 0.641391x^4 + 0.1x^2\,. \tag{36}$$

Assuming a truncated power series for $g(y)$ in Eq. (35), with the corresponding $\tilde{u}(y)$, we obtain

$$\begin{aligned} g(y) = &+ 0.0505146 + 0.451816y^2 + 0.0523064y^4 \\ &+ 0.137535y^6 - 0.140594y^8 + 0.201719y^{10} + O\left(y^{12}\right). \end{aligned}$$

The function $x^6 g(1/x)$ is to be matched to a solution $U_{\text{DS}}(x)$ of Eq. (11) in the form (31). The matching is to be performed at a value of $y$ so small that the $O\left(y^{10}\right)$ expansion is sufficient, and at a value of $x = 1/y$ such that we can also keep few terms of the expansion (31). We choose the value of $x$ that makes the last two terms of the power expansion of $g(y)$ of equal magnitude, namely, $x = 1/y = 1.2$ [at this value, $g(y) = 0.435$ and $0.201719y^{10} = 0.0326$]. Furthermore, we keep in (31) up to $O\left(x^{10}\right)$. After solving the equations for the coefficients, we find

$$\begin{aligned} U_{\text{DS}}(x) = &- 0.00131719 + 0.193442x^2 + 0.341827x^4 \\ &+ 0.143405x^6 - 0.0954937x^8 + 0.0478979x^{10} + O\left(x^{12}\right). \end{aligned} \tag{37}$$

Naturally, this procedure involves numerical errors and the coefficients may not be very precise. The precision can be tested with various numerical experiments. For example, we can (laboriously) include higher powers of $x$ in (31) before solving for the coefficients. If we match $U_{\text{DS}}(x)$ up to $O\left(x^{12}\right)$, then the coefficients change: we observe that the coefficients of powers up to $O\left(x^6\right)$ are quite stable, but the coefficients of $x^8$ and $x^{10}$ become $-0.12$ and $0.17$, respectively. At any rate, they stay small and, therefore, seem to justify the truncation of $U_{\text{DS}}(x)$ at some low order. For instance, we can truncate the expansion (31) at $O\left(x^8\right)$ ($c_5 x^{10} = 0$) and use Eq. (34) with the values of $r$, $\lambda$, and $g$ in (37), to obtain $c_4 = -0.083$, which is surely not very imprecise.

Of course, we can also check on the truncation at $O\left(x^6\right)$ given by Eqs. (24), (25), and (26). For the potential (36), we obtain, by solving these equations, that $r = 0.191997$, $\lambda = 0.347995$, $g = 0.120942$, which agree well with (37). Therefore, the truncation at $O\left(x^6\right)$ works for $r = 0.19$, notably smaller than one. However, let us remark that to assume a truncation at $O\left(x^6\right)$ does not imply to assume that the sextic effective potential is an *exact* solution of Eq. (11), as would occur if Eq. (27) were verified (it is definitely not verified in this case).

We may notice that $r_0 = 0.1$ in (36) has grown to $r = 0.19$ in (37). In fact, $r > r_0$ always, as commented after Eq. (7). Therefore, we need that $r_0 < 0$ to attain $r = 0$. The results that correspond to (36) with $r_0 = -0.1$ instead of $r_0 = 0.1$ are the following: $r = 0.053$, $\lambda = 0.27$, and $g = 0.32$. We see that $r$ is small, so we expect that a more negative value of $r_0$ must make $r \to 0$. Actually, from Eq. (16) we deduce that $r = 0$ when $r_0 = -12\alpha\lambda_0 = -0.194959$. However, we find that the fitting of the power expansion (31) becomes very tricky when we approach this value. We must not be surprised, since we have already seen that then the solution of Eq. (11) is not analytic at $x = 0$ (Sect. 4). However, we still have an acceptable matching of power series for $r_0 = -0.18$ (with $\lambda_0 = 0.641391$ and $g_0 = 0.1$). It yields:

$$\begin{aligned} U_{\text{DS}}(x) = &- 0.00915248 + 0.00585087x^2 + 0.176356x^4 + 1.00058x^6 - 0.705973x^8 \\ &+ O\left(x^{10}\right). \end{aligned} \tag{38}$$

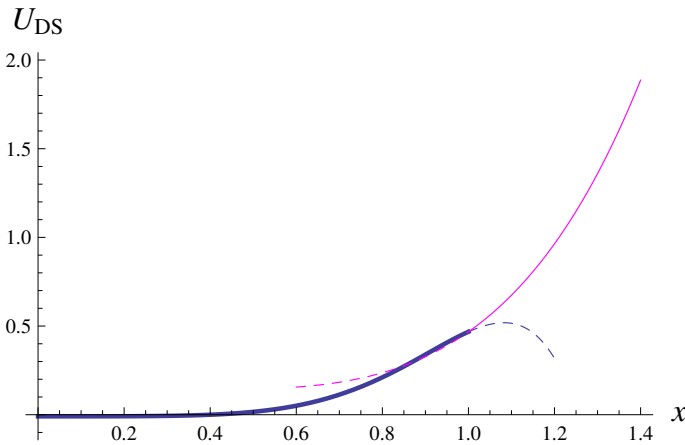

Figure 1: Matching of power series at $x = 1$ for $r = U''_{\mathrm{DS}}(0)/2 = 0.0059$.

Here we have included only up to $O(x^8)$ because higher powers actually worsen the fit, which is a symptom of a reduced radius of convergence. Function (38) is plotted in Fig. 1 together with the matching function $x^6 g(1/x)$. For $r_0$ a bit smaller than $-0.18$, we can still get a sensible power series of $U_{\mathrm{DS}}(x)$, but the matching is progressively worse and the results are less reliable.

It is interesting to note that the truncation at $O(x^6)$ given by Eqs. (24), (25), and (26), agrees well with (38), in spite of $c_4$ being of considerable magnitude. Indeed, by solving the equations, we obtain $r = 0.00584496$, $\lambda = 0.176559$, $g = 0.990523$. This good agreement is due to the small contribution of the term that contains $c_4$ in the right-hand side of Eq. (32), which is therefore almost equivalent to Eq. (26).

## 6 Comparison with other results

Here we compare the foregoing results with other results, encompassing numerical and analytic results. We first compare to Shepard $et\ al$'s numerical work [17]. As regards mass renormalization, Shepard $et\ al$'s Eq. (8) gives Eq. (13), which is unsuitable (Sect. 3.1). However, they actually employ the gap equation (15).

Shepard $et\ al$ set $\lambda_0 = 10$, but their bare coupling $\lambda_0$ is not to be identified with ours, which is divided by $\Lambda_0$ when we undo the change of variables that removes $\Lambda_0$. Furthermore, our normalization of the $\phi^4$-term is different, by a factor of 4. Shepard $et\ al$ show that $\Lambda_0 = (6\pi^2)^{1/3}$ for the lattice calculations in $d = 3$. When both differences are taken into account, we have $\lambda_0 = (10/4)/(6\pi^2)^{1/3} = 0.641391$, the value used in Sect. 5.1. This value is smaller than one but not very small. If we replace $\lambda$ with $\lambda_0$ in Eq. (24) and use this value of $\lambda_0$, then we have a relation between the bare and renormalized values of $r$ (or $m$). For $r \gg 1$, $r$ hardly differs from $r_0$, as expected in the "classical" limit of negligible fluctuations. Shepard $et\ al$ actually set $m = 0.285$ for their "latticized" version of the cactus approximation. This is, with our normalization, equivalent to $m = 0.285/(6\pi^2)^{1/3} = 0.07312 \ll 1$ or $r = m^2/2 = 0.002673$, being in the regime of strong fluctuations. With the ordinary gap equation, we obtain $r_0 = -0.17$, whereas the "latticized" gap equation yields $m_0 = 2.61i$ [17], equivalent to $r_0 = -0.224$. The result of Monte Carlo calculations by Shepard $et\ al$ is $r_0 = -0.154$.

Actually, whether we have $r^{1/2}\lambda_0$, as in the gap equation, or $r^{1/2}\lambda$, as in Eq. (24), is almost irrelevant when $r \ll 1$, because either of them hardly contributes to the right-hand side of each equation. Therefore, $r_0 \approx -12\alpha\lambda_0$ for $r \to 0$. This is $r_0 = -0.19$, for $\lambda_0 = 0.6414$, as already seen in Sect. 5.1. In that section, we also see, in Eq. (38), that we obtain $r > 0.005$ for

$r_0 = -0.18$, which suggests that $-0.19 < r_0 < -0.18$ for Shepard *et al*'s $r = 0.002673$.

In Eq. (24), we also need $\lambda$ to solve for $r_0$, given $r$ and $\lambda_0$. With the truncation at $O(x^6)$, the coupling constant $\lambda$ of the $\phi^4$-theory is given by Eqs. (25) and (26) with $g_0 = 0$ (equations that do not contain $\alpha$). Given $\lambda_0$ and $r$, both these equations constitute a soluble system for $\lambda, g$. Alternatively, we can employ Eq. (32) and Eq. (34) instead of Eq. (26). With $\lambda_0 = 0.6414$ and $r = 0.002673$, both methods do not differ much and yield $\lambda = 0.14$ and $g = 1.4$–1.6. Shepard *et al* obtain (in our normalization of $\lambda$) $\lambda = 0.096$ with Monte Carlo and $\lambda = 0.12$ with their continuous ERG calculation, making numerical fits that assume $g = 0$ (wrongly). In the case of Eq. (38), with the same $\lambda_0$ but $g_0 = 0.1$ and $r = 0.0059$, we have $\lambda = 0.18$ and $g = 1.0$.

Now we quote the ratios $\lambda/m = \lambda/(2r)^{1/2}$, for later use (in Sects. 6.1 and 6.2). Shepard *et al*'s results obtain $\lambda/m = 1.3$ with Monte Carlo and $\lambda/m = 1.7$ with their continuous ERG calculation (with our normalization of $\lambda$), while we obtain $\lambda/m = 1.9$ with truncation and 1.6 with Eq. (38) (for $g_0 = 0.1$). Shepard *et al*'s DS-like 4-point coupling equation [see Eq. (16) or Eq. (38) in Ref. [17]] yields a too low $\lambda/m = 0.57$.

## 6.1 Comparison with perturbation theory

We have considered both perturbative and non-perturbative mass renormalization in Sect. 3.1 and remarked in Sect. 4 the connection of some non-perturbative equations with one-loop renormalized perturbation theory. It behooves us to proceed to higher orders in the renormalized perturbation series.

Renormalization of the $\phi^4$-theory at the two-loop level is described in detail by Amit, Ref. [42, p. 117 ff]. Applying Amit's formulas to $d = 3$ with momentum cutoff regularization and neglecting inverse powers of the cutoff $\Lambda_0$, the result is:

$$m_0^2 = m^2 - \frac{6\Lambda_0}{\pi^2}\lambda + \frac{3m\lambda}{\pi} - 27\left(\frac{\Lambda_0}{\pi^3 m} - \frac{8+\pi^2}{2\pi^4}\right)\lambda^2 + \frac{6\lambda^2}{\pi^2}\left(\log\frac{\Lambda_0}{m} - 1.5247\right), \qquad (39)$$

$$\lambda_0 = \lambda + \frac{9\lambda^2}{2\pi m} + \frac{63\lambda^3}{4\pi^2 m^2}. \qquad (40)$$

(The last parenthesis in Eq. (39) results from the calculation of the cutoff "sunset" Feynman graph, whose $\Lambda_0$-independent part is computed numerically). These two equations can also be obtained as particular cases ($g_0 = 0$) of equations derived in the appendix.

In the mass renormalization equation (39), the three initial terms in the right-hand side form the one-loop level result, matching equation (13). This equation fails in the massless limit, as analyzed in Sect. 3.1. The addition of the two-loop terms does not remedy it, because they also vanish as $m \to 0$, given that $\lambda/m$ stays finite.

The renormalization of $\lambda$ is given by Eq. (40). This equation, as well as Eq. (39), is fine as an expansion in powers of $\lambda$ for fixed $m$, that is to say, is fine for $\lambda/m$ small. For example, in potential (37), $\lambda/m = 0.55$ and

$$\frac{9\lambda}{2\pi m} = 0.79, \quad \frac{63\lambda^2}{4\pi^2 m^2} = 0.48.$$

Consequently, the power series expansion seems to show certain degree of convergence in this case and higher-order corrections should improve the result (in fact, these series are known to be asymptotic and not convergent [37, 38], and only a limited number of terms can be used). Given $\lambda_0$ and $m$, we could solve for $\lambda$ and obtain an approximate value.

The potential with $r = 0.053$ and $\lambda = 0.27$ in Sect. 5.1 gives $\lambda/m = 0.83$, larger than for potential (37), and

$$\frac{9\lambda}{2\pi m} = 1.2, \quad \frac{63\lambda^2}{4\pi^2 m^2} = 1.1.$$

Naturally, cases with $\lambda/m \gtrsim 1$ are unsuitable (we have found above that $\lambda/m > 1$ with Shepard *et al*'s values of $\lambda_0$ and $m$). In these cases, the raw form of $\lambda_0/m$ as a series expansion in powers of $\lambda/m$ is hardly useful and series summation methods are the standard procedure [37, 38].

The perturbative expansion of the effective potential of the $\phi^4$-theory in $d = 3$ is known to the five-loop order [23]. However, the calculations have been done in dimensional regularization with minimal subtraction, instead of being done with a physical momentum cutoff. Dimensional regularization introduces an unknown scale $\mu$ and misses any power of $\Lambda_0$. The redefinition to the physical mass scheme is a very laborious task [23]. On the other hand, the calculation of Feynman graphs with a momentum cutoff is also very laborious.

Perturbative $\phi^6$-theory is even more complicated, of course. Although its study was initiated long ago [43–45], there are only very partial results. While perturbative $\phi^4$-theory is *super-renormalizable* in $d = 3$, $\phi^6$-theory is just renormalizable and divergences proliferate, according to the general theory [38]. For example, the coupling constant $\lambda$ is divergent in the limit $\Lambda_0 \to \infty$; say, it is non universal [as already manifest in Eq. (25)]. Perturbative computations with dimensional regularization and minimal subtraction by McKeon and Tsoupros [46, 47] or Huish and Toms [48, 49] are hardly useful in our context, because they are restricted to the poles in $\varepsilon = 3 - d$ and omit the finite parts. Much the same can be said of the six-loop calculation of Hager [50], which applies strictly to tricritical behavior. Lawrie and Sarbach discuss this type of renormalization and its limitations [29, Sect. V]. Sokolov [45] calculated in the physical mass scheme but only retained quadratic terms in the coupling constants (as do McKeon and Tsoupros [46, 47], while Huish [49] has completed a four-loop order calculation). In fact, Sokolov's [45] non-trivial RG fixed point (the Wilson-Fisher fixed point) is not correct: at that fixed point, $g = 0$, but later calculations contradicted it [22, 23].

We present in the appendix the full two-loop renormalization of $(\lambda\phi^4 + g\phi^6)_3$ theory, regularized with the momentum cutoff $\Lambda_0$ in the physical mass scheme. The comparison between the non-perturbative equations and the perturbative ones shows that the latter perform worse. For example, the results of Eqs. (24), (25) and (26) for potential (36) reproduce the full non-perturbative potential (37) within a few percent in the case of $r$ and $\lambda$, while the one-loop perturbative results, namely, $r = 0.122808$, $\lambda = 0.338735$, $g = 0.172128$, deviate more. Of course, the exact results are not available and we can only try to draw conclusions by comparing between various results.

We find, furthermore, that the two-loop contribution actually spoils the one-loop approximation obtained for potential (36). In fact, it is difficult to find numerical solutions for the renormalized coupling constants from Eqs. (A.8), (A.9), and (A.10), which are very complex. If we consider only the perturbation series for $g(m, \lambda, g_0)$ in Eq. (A.7), then we see that it appears to be strongly non-convergent, as manifested in a negative and relatively large two-loop contribution to $g$, when we put $g_0 = 0.1$, $m = \sqrt{2r} = 0.622$ and $\lambda = 0.3418$ [Eq. (37)]. In the simple case with $g_0 = 0$, Eq. (A.7) simplifies to

$$g(m, \lambda, 0) = \frac{9\lambda^3}{\pi m^3}\left(1 - \frac{3\lambda}{\pi m}\right) + O\left(\lambda^5\right), \tag{41}$$

as previously derived by other authors [22, 23], who noticed that these expansions are not useful for $\lambda/m$ close to one. The expansions actually get worse at higher loop order and demand resummation methods [22, 23, 26] (unless $\lambda/m$ is small, of course).

The convergence of series expansions with $g_0 \neq 0$ is worse than with $g_0 = 0$, and no resummation method has ever been tried in that case. In fact, we can see in the series for $g(m, \lambda, g_0)$ of Eq. (A.7) that the term with $\log(m/\Lambda_0)$ will grow without bound when $m \to 0$. This growth is harmless for tricritical behavior and actually gives the expected logarithmic correction to scaling behavior [43–45]. However, that term shows the problems that arise to reach ordinary critical behavior by using perturbation theory when $g_0 \neq 0$. One obvious

problem arises in the $\varepsilon$-expansion, because it requires a different dimension base for either tricritical behavior or critical behavior, as discussed by Lawrie and Sarbach [29, Sect. V].

## 6.2 The massless limit: critical behavior

Our approach leads us to consider, in $d = 3$, the $\phi^6$-theory instead of just the $\phi^4$-theory. The result of the perturbative renormalization group [43–45] is that $g\phi^6$ is RG-irrelevant with respect to the trivial RG fixed point, which corresponds to tricritical behavior ($\lambda/m = 0$). Of course, $g\phi^6$ is also irrelevant with respect to the non-trivial fixed point which corresponds to ordinary critical behavior ($\lambda/m \neq 0$). Thus, one does not need to take $g_0 = 0$ to study the critical behavior, and the renormalized value of $g$ should be independent of $g_0$. In contrast, $\lambda$ is irrelevant with respect to the non-trivial fixed point but is relevant with respect to the trivial fixed point. In the critical surface, there is a RG trajectory that departs from the trivial fixed point and leads to the non-trivial fixed point. The higher stability of the latter implies that the massless theory will be controlled by this fixed point, for generic values of the bare couplings [1, 28, 29, 45]. We have seen in Sect. 3.1 that our approach is adequate for mass renormalization in the massless limit. Here we consider coupling-constant renormalization.

Our coupling constants $g$ and $\lambda$ are already dimensionless, since $\lambda$ is the dimensionful coupling constant divided by $\Lambda_0$. However, it is convenient here to define a quartic coupling constant that is dimensionless and does not involve $\Lambda_0$, as is common; namely,

$$u = \lambda/m = \lambda/(2r)^{1/2}.$$

In terms of this variable, Eq. (27), for example, adopts the form

$$g = 6u^2/5, \tag{42}$$

which is non-perturbative, unlike Eq. (41), but is not exact (note that the two equations do not agree for $u \ll 1$).

Let us recall how the RG fixed point value $u^*$ that controls the critical behavior is derived in perturbative $\phi^4$-theory at fixed dimension [37, 38]. While ordinary perturbation theory yields $u(m, \lambda_0, \Lambda_0)$ as a power series in $\lambda_0$, renormalized perturbation theory, in $d = 3$, simply yields $\lambda_0/m$ as a power series in $u$ [e.g., Eq. (40)]. This expansion can be reversed to obtain the expansion of $u(\lambda_0/m)$. The value $u^*$ is determined by the vanishing $\beta$-function condition

$$\left(\frac{\partial u}{\partial m}\right)_{\lambda_0} = 0. \tag{43}$$

In fact, $\lambda_0/m$ as a function of $u$ should have a vertical asymptote at $u^*$, such that the solution of Eq. (43) is found for $m \to 0$ and $\lambda_0/m \to \infty$ (as explained by Parisi [37, §8.1]). The vertical asymptote at $u^*$ corresponds to a horizontal asymptote of $u(\lambda_0/m)$, that is to say, to a maximum value of $u$ for $m \to 0$ at fixed $\lambda_0$. However, any truncated perturbative series of $\lambda_0/m$ in powers of $u$ is a polynomial and does not have a vertical asymptote [e.g., Eq. (40)]. Nevertheless, if one *assumes* that one can obtain $\beta(u)$ as a power series expansion truncated at the same order, then one can solve the equation $\beta(u^*) = 0$ and determine $u^*$. Of course, this value necessarily corresponds to a finite value of $\lambda_0/m$ and hence to $m > 0$. Moreover, the expansion of $u(\lambda_0/m)$, truncated at the same order, may or may not have a maximum [the inverse expansion of (40) does not]; and if it does have a maximum, it may not be at the value $u^*$ determined by $\beta(u^*) = 0$.

The above arguments show that the perturbative calculations of critical behavior are questionable. At low orders, the RG fixed point is found at $\lambda_0/m \sim 1$ rather than $\lambda_0/m \gg 1$. Nevertheless, $\lambda_0/m$ grows as higher-order terms are added, that is to say, as higher powers of

$\lambda$ are added to the right-hand side of Eq. (40). In fact, perturbation theory at fixed dimension can be arranged to yield good results at higher orders [37, 38]. At any rate, the higher-order terms make a substantial contribution, as already seen for moderately small $m$ in Sect. 6.1. The calculation of $u^*$ demands sophisticated series summation methods (described by Zinn-Justin in Ref. [38, ch. 42]).

Moreover, renormalized perturbation theory is not useful for knowing whether or not $m = 0$, given the initial values $m_0$ and $\lambda_0$. The problem with the massless limit has already been presented in Sect. 3.1. Indeed, this problem is traditionally treated non-perturbatively, for example, with the use of the gap equation.

Let us now consider various results of perturbative and non-perturbative calculations of $u^*$ and $g^*$ (the critical sextic coupling constant). Work done years ago by Tsypin [18] on the effective potential of the 3d Ising universality class, ruled by the $\phi^4$-theory, employing Monte Carlo simulations, yielded $u^* = 0.97 \pm 0.02$ and $g^* = 2.05 \pm 0.15$. The consensus value is $u^* \simeq 1$, according to Tsypin [18]. Zinn-Justin reports (a value equivalent to) $u^* = 0.985$ in Ref. [38, table 29.4], obtained with perturbative calculations (see also Ref. [25, table 8]). The values of $u$ found by Shepard $et$ $al$ and by our approach close to the massless limit are somewhat larger: Shepard $et$ $al$'s Monte Carlo calculation gives $u = 1.3$ for $r = 0.0027$ and our approach gives $u = 1.6$ for $r = 0.0059$. The latter actually comes from $\lambda\phi^4 + g\phi^6$ theory, but $u^*$ should be the same as in just the $\phi^4$ theory.

The sixth degree coupling constant $g$ is also interesting. Tsypin [18] referred to results of perturbation theory in fixed dimension $d = 3$ as well as results of the $\varepsilon$-expansion. He also referred to results of the ERG in the local potential approximation. In summary, the quoted results show that $g^* \in (1.6, 2.3)$. Guida and Zinn-Justin [23] or Sokolov and collaborators, in Ref. [22] and later in Ref. [26], with a more modern treatment, report $g^* = 1.65$. If $u^* \simeq 1$ and Eq. (42) hold, then we would have that $g^* \sim 6/5 = 1.2$. Butera and Pernici [19, table X] compare various results, including higher-order coupling constants. The variance of $g^*$ in those results is considerable. Higher order coupling constants have larger errors. We have found, for $\lambda_0 = 0.6414$ and $r = 0.002673$, with the truncated non-perturbative potential, $g = 1.4$–$1.6$, whereas the asymptotic matching technique gives $g = 1.0$ in (38) for a somewhat larger $r$.

## 7   Discussion

We have explored the connection between bare and renormalized couplings that is derived from the integral form of the Wilson or exact renormalization group. Our approach is a generalization of Shepard $et$ $al$'s integral equation and consists in a sort of a two-step rather than a one-step integration over the momentum scale. It gives rise, in three dimensions, to a second-order differential equation for the effective potential, namely, Eq. (11). Of course, a many-step integration is possible, but it involves many functions and boils down to a procedure of numerical integration of the ERG by discretization of $\Lambda$. In contrast, Eq. (11) can be studied analytically and gives insight into non-perturbative renormalization.

The basic issue to be discussed is the scope of differential equation (11). To evaluate it, we have first found an exact solution and then studied the general solution (in three dimensions) in its light. In addition, we compare our numerical results to results of other approaches, mainly with the results of Shepard $et$ $al$ and, from a different perspective, with the results of perturbation theory.

In the simple case of mass renormalization, we have compared Eq. (11) with three other equations for the effective potential, namely, the perturbative one-loop equation, Shepard $et$ $al$'s Eq. (8), and the gap equation. While Shepard $et$ $al$'s equation totally fails in the massless limit, Eq. (11) and the gap equation, which both approximate the Dyson-Schwinger equation

for the two point function, correctly reproduce the expected result in the massless limit (for small values of $m_0$ and $\lambda_0$). Moreover, the mass renormalization equation that derives from Eq. (11), namely, Eq. (14), takes into account that the quartic coupling constant is renormalized as well, as occurs in the Dyson-Schwinger equations but is neglected in the gap equation.

The renormalization of mass involves the renormalized quartic coupling constant and, likewise, the renormalization of any coupling constant involves the next-order renormalized coupling constant. Therefore, our approach, when stated as a set of equations for the renormalization of coupling constants, needs a truncation, as do the Dyson-Schwinger equations; that is to say, we face an infinite tower of equations that need to be truncated. The truncation up to the sextic coupling works well, as shown in Sect. 6. At any rate, our crucial result is that the indeterminacy implicit in the infinite tower of equations corresponds to the indeterminacy of the initial value problem for the second order differential equation. In this formulation of the problem, we can obtain definite results by appealing to the asymptotic behavior of the general solution of the differential equation, namely, by demanding that the solution has no singular behavior for large values of the field.

The necessity of a boundary condition at infinity is surely not a particular feature of our simplification of the ERG equation for the effective potential. Arguably, the absence of such boundary condition has hampered attempts at numerical integration of the ERG equation. In this regard, we expect that the insight provided by our simplified approach constitutes a step forward towards successful methods of integration of the exact equation.

An essential innovation in our approach is that it naturally leads us to consider, in $d = 3$, the renormalizable $\phi^6$-theory rather than the super-renormalizable $\phi^4$-theory. The standard treatments of the ERG in three dimensions are compared to results of perturbative $\phi^4$-theory, because the focus is on the *non-trivial* RG fixed point, which can be found with just the $\phi^4$-theory. However, there are interesting phenomena related to the trivial (tricritical) fixed point and, especially, to the crossover from one point to another. These phenomena have not been considered within a non-perturbative approach, but we have shown that it is natural to do so. This is an aspect to develop in future non-perturbative studies of scalar field theory in three dimensions.

On the other hand, $(\lambda \phi^4 + g \phi^6)_3$ theory can also be studied with perturbative methods, but it is quite complicated and our non-perturbative approach seems definitely superior. To better answer this question, we should have more extensive results of the perturbative method than those currently available.

The massless limit of the effective potential deserves a specific discussion. It is troublesome, in perturbation theory as well as in our non-perturbative approach, because the massless (or critical) theory should have a non-analytic effective potential. We have seen that the differential equation for the effective potential, in our approach, is indeed singular in the massless limit. Actually, the singularity does not prevent us from carrying out a numerical integration of the differential equation. However, we have found that the nature of the singularity is not easily established by numerical methods. Further investigation of this question is left for the future.

Finally, let us remark that convexity of the effective potential is required to have a well defined 3/2-power in our differential equation for the potential. In the consequent relation between classical and renormalized parameters, we see that there appears $r^{1/2} = m$ (besides the possibly negative $r_0 = m_0^2$), and $m$ becomes an imaginary number in a broken symmetry phase. The cure for this problem is well-known: to redefine the mass by a shift of the field so that the expansion of the potential is made about a true minimum. Likewise, further elaboration of our method is required to extend it to broken symmetry phases. This extension should present no special difficulties, because it has been shown that the ERG directly obtains a convex potential in broken symmetry phases. In any case, our approach describes

the non-perturbative effect of quantum corrections on the onset of symmetry breaking when the parameters of the classical potential are tuned to the massless limit. In particular, we have shown that there is a definite improvement in the mass renormalization equation in that limit.

## Acknowledgments

I would like to thank Sergey Apenko for comments on the manuscript.

## A  Two-loop perturbative $(\lambda\phi^4 + g\phi^6)_3$ theory in the physical mass scheme

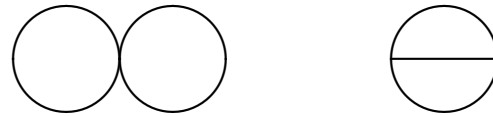

The effective potential at two-loop order is calculated with the background-field method (succinctly exposed by Honerkamp [51]):

$$
U_{\text{eff}}(x) = U_{\text{clas}}(x) + h\left(\frac{\Lambda_0 U''_{\text{clas}}(x)}{4\pi^2} - \frac{U''_{\text{clas}}(x)^{3/2}}{12\pi}\right) + \frac{3h^2 U^{(4)}_{\text{clas}}(x)}{4!}\left(\frac{\Lambda_0^2}{4\pi^4} - \frac{\Lambda_0\sqrt{U''_{\text{clas}}(x)}}{4\pi^3}\right.
$$
$$
\left. + \frac{(8+\pi^2)U''_{\text{clas}}(x)}{16\pi^4}\right) - \frac{3h^2 U^{(3)}_{\text{clas}}(x)^2}{(3!)^2(16\pi^2)}\left(A - \log\frac{\sqrt{U''_{\text{clas}}(x)}}{\Lambda_0}\right) + O(h^3)\,. \qquad \text{(A.1)}
$$

Here, the loop counting parameter is $h$. The $O(h)$ term is standard. There are two $O(h^2)$ terms, given by the "double bubble" and the "sunset" Feynman graphs, drawn above. The corresponding integrals are calculated with momentum cutoff $\Lambda_0$ (the finite part of the "sunset" graph integral is computed numerically, yielding $A = -1.5247$). Both $h$ and $\Lambda_0$ are to be set to 1 at the end, to compare with the non-perturbative potential $U_{\text{DS}}(x)$. Of course, we put

$$
U_{\text{clas}}(x) = \frac{m_0^2}{2}x^2 + \lambda_0 x^4 + g_0 x^6,
$$

in accord with Eq. (21) (the constant term can be suppressed). Let us remark that, if we make $g_0 = 0$, then we reproduce the results in several references (calculated at higher loop order) [20–25].

The renormalization is carried out as follows:

$$
m^2 = U''_{\text{eff}}(0) = m_0^2 + h\left(\frac{6\lambda_0\Lambda_0}{\pi^2} - \frac{3m_0\lambda_0}{\pi}\right) + h^2\left[\frac{45g_0\Lambda_0^2}{2\pi^4} - \frac{45g_0 m_0\Lambda_0}{2\pi^3} + \left(\frac{45}{8\pi^2} + \frac{45}{\pi^4}\right)g_0 m_0^2\right.
$$
$$
\left. - \frac{9\Lambda_0\lambda_0^2}{\pi^3 m_0} + \left(\frac{9}{2\pi^2} + \frac{36}{\pi^4} - \frac{6A}{\pi^2}\right)\lambda_0^2 + \frac{6\lambda_0^2}{\pi^2}\log\frac{m_0}{\Lambda_0}\right]\,, \qquad \text{(A.2)}
$$

$$\lambda = \frac{U_{\text{eff}}^{(4)}(0)}{4!} = \lambda_0 + h\left(-\frac{15g_0 m_0}{4\pi} + \frac{15g_0\Lambda_0}{2\pi^2} - \frac{9\lambda_0^2}{2\pi m_0}\right) + h^2\left[-\frac{315\Lambda_0 g_0\lambda_0}{4\pi^3 m_0}\right.$$
$$\left. + \left(\frac{315}{8\pi^2} + \frac{315}{\pi^4} - \frac{30A}{\pi^2}\right)g_0\lambda_0 + \frac{30g_0\lambda_0}{\pi^2}\log\frac{m_0}{\Lambda_0} + \frac{27\lambda_0^3\Lambda_0}{2\pi^3 m_0^3} + \frac{18\lambda_0^3}{\pi^2 m_0^2}\right], \quad \text{(A.3)}$$

$$g = \frac{U_{\text{eff}}^{(6)}(0)}{6!} = g_0 + h\left(\frac{9\lambda_0^3}{\pi m_0^3} - \frac{45g_0\lambda_0}{2\pi m_0}\right) + h^2\left[-\frac{675g_0^2\Lambda_0}{4\pi^3 m_0} + \left(\frac{675}{8\pi^2} + \frac{675}{\pi^4} - \frac{75A}{\pi^2}\right)g_0^2\right.$$
$$\left. + \frac{75g_0^2}{\pi^2}\log\frac{m_0}{\Lambda_0} + \frac{270g_0\lambda_0^2\Lambda_0}{\pi^3 m_0^3} + \frac{225g_0\lambda_0^2}{\pi^2 m_0^2} - \frac{81\lambda_0^4\Lambda_0}{\pi^3 m_0^5} - \frac{108\lambda_0^4}{\pi^2 m_0^4}\right]. \quad \text{(A.4)}$$

[The symbol $O(h^3)$ that denotes higher order terms is omitted in some expressions, for brevity.]

Solving for $m_0$ in Eq. (A.2) and substituting in the following two equations:

$$\lambda = \lambda_0 - h\,\frac{3\left(5\pi g_0 m^2 - 10g_0 m\Lambda_0 + 6\pi\lambda_0^2\right)}{4\pi^2 m} + h^2\left[-\frac{135\Lambda_0 g_0\lambda_0}{2\pi^3 m} + \left(\frac{135}{4\pi^2} + \frac{315}{\pi^4} - \frac{30A}{\pi^2}\right)g_0\lambda_0\right.$$
$$\left. + \frac{30g_0\lambda_0}{\pi^2}\log\frac{m}{\Lambda_0} + \frac{99\lambda_0^3}{4\pi^2 m^2}\right] + O(h^3), \quad \text{(A.5)}$$

$$g = g_0 - h\,\frac{9\left(5g_0 m^2\lambda_0 - 2\lambda_0^3\right)}{2\pi m^3} + h^2\left(-\frac{675g_0^2\Lambda_0}{4\pi^3 m} + \frac{675g_0^2}{8\pi^2} + \frac{675g_0^2}{\pi^4} - \frac{75Ag_0^2}{\pi^2} + \frac{75g_0^2}{\pi^2}\log\frac{m}{\Lambda_0}\right.$$
$$\left. + \frac{405g_0\lambda_0^2\Lambda_0}{2\pi^3 m^3} + \frac{1035g_0\lambda_0^2}{4\pi^2 m^2} - \frac{297\lambda_0^4}{2\pi^2 m^4}\right) + O(h^3). \quad \text{(A.6)}$$

Hence, we obtain $g(m, \lambda, g_0)$ as:

$$g = g_0 - h\,\frac{9\left(5g_0 m^2\lambda - 2\lambda^3\right)}{2\pi m^3}$$
$$+ h^2\left(\frac{675g_0^2}{\pi^4} - \frac{75Ag_0^2}{\pi^2} + \frac{75g_0^2}{\pi^2}\log\frac{m}{\Lambda_0} + \frac{1035g_0\lambda^2}{4\pi^2 m^2} - \frac{27\lambda^4}{\pi^2 m^4}\right) + O(h^3). \quad \text{(A.7)}$$

We can also solve for $m_0$, $\lambda_0$ and $g_0$ in Eqs. (A.2), (A.3) and (A.4):

$$m_0^2 = m^2 + h\,\frac{3\left(\pi m\lambda - 2\lambda\Lambda_0\right)}{\pi^2} + h^2\left[\frac{45g\Lambda_0^2}{2\pi^4} - \frac{45gm\Lambda_0}{2\pi^3}\right.$$
$$\left. - \frac{27\lambda^2\Lambda_0}{\pi^3 m} + \left(\frac{27}{2\pi^2} - \frac{36}{\pi^4} + \frac{6A}{\pi^2}\right)\lambda^2 - \frac{6\lambda^2}{\pi^2}\log\frac{m}{\Lambda_0} + \left(\frac{45}{8\pi^2} - \frac{45}{\pi^4}\right)gm^2\right], \quad \text{(A.8)}$$

$$\lambda_0 = \lambda + h\,\frac{3\left(5\pi gm^2 - 10gm\Lambda_0 + 6\pi\lambda^2\right)}{4\pi^2 m} + h^2\left(-\frac{675\Lambda_0 g\lambda}{4\pi^3 m}\right.$$
$$\left. + \frac{675g\lambda}{8\pi^2} - \frac{315g\lambda}{\pi^4} + \frac{30Ag\lambda}{\pi^2} - \frac{30g\lambda}{\pi^2}\log\frac{m}{\Lambda_0} + \frac{135\Lambda_0\lambda^3}{2\pi^3 m^3} - \frac{18\lambda^3}{\pi^2 m^2}\right), \quad \text{(A.9)}$$

$$g_0 = g + h\,\frac{9\left(5gm^2\lambda - 2\lambda^3\right)}{2\pi m^3}$$
$$+ h^2\left(-\frac{675g^2}{\pi^4} + \frac{75Ag^2}{\pi^2} - \frac{75g^2}{\pi^2}\log\frac{m}{\Lambda_0} + \frac{495g\lambda^2}{2\pi^2 m^2} - \frac{351\lambda^4}{2\pi^2 m^4}\right). \quad \text{(A.10)}$$

Replacing these bare constants in $U_{\text{clas}}$ and substituting in Eq. (A.1), we can derive the renormalized expansion of $U_{\text{eff}}$ to $O(h^2)$, in which the limit $\Lambda_0 \to \infty$ obtains a universal potential. Eqs. (A.8) and (A.9) are also useful to derive Eqs. (39) and (40), by replacing in Eqs. (A.8) and (A.9) the function $g(m, \lambda, 0)$, as given by Eq. (A.7) [only needed to $O(h)$].



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
