# Peer review of "Renormalization group and effective potential: a simple non-perturbative approach"

_SciPost Physics, doi:SciPost Phys. Core 5, 044 (2022)_

## Round 2 · Referee Report · Anonymous (Referee 1) · 2022-5-17

Strengths

1- Clear language; 2- Detailed computations; 3- Improvement on previous results from the literature; 4- Comparison between the results obtained in the paper with previous results from literature.

Weaknesses

1- It is not presented a deeper study of the tricritical behavior of the theory; 2- The author doesn't adress theories with spontaneously bronken symmetries; 3- It is not discussed possible implications in adding more fields to the model.

Report

The paper adress a interesting topic and presents an improvement in computing the nonperturbative effective potential. This is of interest to the field in question. Yet there are some questions that need to be clarified before publication.

The first is regard to the substitution of the integration variable by the lower integration limit in equation (5). The author argues that if one uses this limit, it will obtain the desired result while if one uses the upper integration limit it will obtain the one-loop aproximation. This is crucial to the paper since it consists of a improvement of this method.

Secondly, the author discusses the massless limit of this model and states that it is problematic using one method and not using the one developed in the paper. However, one expects that the computation of the effective potential for a classical massless theory dinamically breaks the symmetry, see (Sidney Coleman and Erick Weinberg, Phys. Rev. D 7, 1888 (1973)), (Steven Weinberg
Phys. Rev. D 13, 974 (1976)), (D. G. C. Mckeon
International Journal of Theoretical Physics volume 37, pages817–826 (1998)), (Huan Souza, L. Ibiapina Bevilaqua, and A. C. Lehum
Phys. Rev. D 102, 045004 (2020)). It would be interesting put into perspective the results from dynamical symmetry breaking and the ones developed in the paper.

Moreover, the paper is well written and gives a positive contribution to ongoing discussions of the field.

Requested changes

1- Needs clarifications about the procedure to identify equation (5) as equation (2).

2- It would be nice to discuss if it is possible to have dynamical symmetry break using this method and put the results from this method into perspective with the ones obtained via perturbation theory.

---

## Round 2 · Referee Report · Anonymous (Referee 2) · 2022-5-25

Weaknesses

1- Incremental study of an arbitrary equation for the effective potential 2- No clear goal and motivation 2- Reference to the numerous studies of LPA missing

Report

This manuscript is focused on the approximate solution of an Exact Renormalization Group (ERG) equation at the Local Potential Approximation (LPA), see Eq. (3). The latter is a PDE for the effective potential, which depends on the "classical field" phi and an RG scale Lambda.
Twenty-five years ago, Shepard et al. gave an approximate solution to this ERG equation to obtain a self-consistent equation for the renormalized (Lambda=0) effective potential.
Here, the author gives another approximate solution by splitting the integration range in two, and replacing in an ad hoc way the flowing potential by either its classical value (at the UV scale Lambda_0) or its renormalized value. After some more approximations, Eq. (11) is obtained and then studied at length in the rest of the paper.

My main issue with the manuscript is that there is no clear motivation as to why this should be an interesting equation to study, and what new information we can gain from it. Indeed, in the end, and as stated by the author himself in the conclusion, Eq. (11) is just an approximate solution to the LPA flow equation. Then, why not just study that equation? One issue of course is that the LPA has been studied at length in the last thirty years, with a variety of regulator functions, and therefore it is hard to see what new insight we would get from yet another study of the LPA.

Furthermore, the "difficulties" listed in the conclusion, such as "the requirement of convexity of the effective potential prohibits the study of symmetry broken phase" and "The massless limit of the effective potential [...] is troublesome [...] because the massless (or critical) theory should have a non-analytic effective potential. " are very well understood in the ERG context.
However, to correctly capture the behavior at criticality (non-analyticity) and in the broken-symmetry phase, it is well-known that one needs to: 1) be functional (i.e. not expand the effective potential); 2) solve the RG flow (i.e. keep a flowing effective potential in the RHS of the flow equation). Among the many possible references on the second aspect: Nuclear Phys. B 855 (2012) 854, Phys. Rev. E 94, 042136 (2016).

To summarize, this manuscript studies in detail an equation that is not able to capture some phenomena that are very well understood in the context of ERG (and leave that for future work). As such, I do not see how it could be published without major revisions and a thorough study of the literature.

---

## Round 2 · Referee Report · Anonymous (Referee 3) · 2022-6-20

Strengths

1 - Well written.
2 - Thorough comparison with relevant literature.
3 - Explicit computations.

Weaknesses

1 - Lacking clear motivation for the main object under study.
2 - Manipulations leading to the main equation under study require additional substantiation.
3 - The utility of the method to future applications is unclear.

Report

The author considers an effective potential $U_{DS}$ introduced by Shepard et al, which is an approximate solution to the exact RG and satisfies a second order differential equation. They propose a two-step integration procedure to approximate the solution to this equation, and demonstrate that implementing this procedure reduces error. They then proceed to analyze the resulting effective potential.

The author convincingly makes the point that their two-step integration procedure is an improvement on the one-step procedure, providing concrete calculations to support this claim.

However, this paper would greatly benefit by establishing a firmer initial motivation for analyzing $U_{DS}$. This object was introduced by Shepard et al as a “crude integration of the ERG equation” (see page 5 of the paper under review), which itself is an approximation of the effective potential to lowest order in the derivative expansion. In this reviewer’s opinion, the author should more convincingly address why this particular approximation is interesting and worth detailed study.

Relatedly, it would be helpful to have a clearer discussion of the sense in which this$~U_{DS}$ is approximate, since this point is so central to the paper. The author discusses the manner of deviation of equation (2) from the sharp cutoff ERG equation (equations (3)-(5) in various representations), but the effect of the deviation is unclear. Is there (for instance) an estimate for the error accrued in making the assumption that one can replace$~U’’(\phi,k)$ in equation (5) with its value at the lower integration limit?

Furthermore, this reviewer does not understand the author’s claim that the “requirement of convexity of the effective potential prohibits the study of symmetry broken phases”. The study of spontaneous symmetry breaking in the context of (e.g.) the derivative expansion of the effective potential has a long and famous history, going back to the work of Coleman and Weinberg. Even if spontaneous symmetry breaking is not studied in the current paper, the discussion should address how one might do so in the future.

Requested changes

1 - Clarify the motivations for a detailed study of $U_{DS}$.
2 - Explain the error in going from equation (5) to equation (2).
3 - Comment on the application to spontaneous symmetry breaking.

---

## Round 3 · Referee Report · Anonymous (Referee 1) · 2022-7-6

Strengths

1- Clear language; 2- Detailed computations; 3- Improvement on previous results from the literature; 4- Comparison between the results obtained in the paper with previous results from literature.

Weaknesses

1 - It could adress richer models, e.g. with more fields; 2- Does not provide explicity calculations for models with spontaneously broken symmetries.

Report

The author answered the points made in my last report. Although it was not explicitly included computations for a model with broken symmetry, the paragraphs introduced in the paper sufficiently explain what one may expect from what would happen when studying such models. I understand that a detailed computation on this would require a lot of work.

The paper meets the acceptance criteria "Open a new pathway in an existing or a new research direction, with clear potential for multipronged follow-up work". That is because it introduces an improved way to compute the effective potential and compare its results with previous ones from the literature, showing that there is a numerical correction. This can be crucial when studying critical points (as discussed by the author in the paper) as well as vacuum stability, as mentioned in Degrassi, G., Di Vita, S., Elias-Miró, J. et al. Higgs mass and vacuum stability in the Standard Model at NNLO. J. High Energ. Phys. 2012, 98 (2012).

Therefore, I recommend the publication of the paper.
  • validity: good
  • significance: good
  • originality: ok
  • clarity: high
  • formatting: excellent
  • grammar: perfect

Author:  Jose Gaite  on 2022-07-16  [id 2665]

(in reply to Report 1 on 2022-07-06)

I thank Reviewer 1 for his/her positive report and for his/her recommending that I address richer models, with several fields. I have downloaded J. High Energ. Phys. (2012) 98. The standard model is no easy theory. I would like first to consider theories with discrete symmetries, which are more amenable. But I will keep in mind the recommended subject.

---

## Round 3 · Referee Report · Anonymous (Referee 2) · 2022-7-11

Weaknesses

1- A clear motivation for the study of the ad hoc equation is still missing

Report

Reading in detail the other reports and the reply from the author, I must say that I am still entirely convinced that this manuscript cannot be published as it is.
Even in its new version, the motivation of this work is still very weak, as it is still unclear as to why one should be interested in the approximate study of Eqs. (3) or (5).

Indeed, contrary to what the author claims in his reply to Report 1, these two equations are NOT exact formulations of the ERG! They are, as rightly stated in Report 3, just an approximation of the exact equation of the effective potential at the lowest order of the derivative expansion (i.e. the LPA), using an ultra-sharp regulator function.
The author has not given any convincing reasons why it is interesting to solve approximately (even if it is better than how Sherpard et al did 35 (!!!) years ago).

There are however good reasons not to use this regulator: for instance, it prevents going beyond the LPA (say, to the second-order of the derivative expansion), which one should hope to do to obtain better results. So it is hard to see what is gained in focusing so much on that particular equation, and not say the LPA in general, the study of which has been thorough starting from the 90’s.
How does the author plan to improve his method beyond the LPA?

Furthermore, I still do not see what new insight has been gained from this approximate solution of this approximate equation.
What have we learned that was not known before?
Furthermore, what is left for “future work” is also well understood in the context of the ERG. What will be gained from this approach? In my opinion, not much. It might be a “new path”, but it is wandering in a well-known landscape and is not shedding any new or insightful light on these old problems.

Finally, I would also like to point out that writing “The necessity of a boundary condition at infinity is surely not a particular feature of our simplification of the ERG equation for the effective potential. Arguably, the absence of such boundary conditions has hampered attempts at numerical integration of the ERG equation. In this regard, we expect that the insight provided by our simplified approach constitutes a step forward towards successful methods of integration of the exact equation.” is highly misleading, and it should be corrected. The ERG equations have been integrated numerically for decades now (using more advanced approximations and for much more complicated problems than a scalar theory), without major difficulties coming from the boundary at infinity. This can also be dealt with by compactifying the field for instance.
  • validity: ok
  • significance: poor
  • originality: low
  • clarity: ok
  • formatting: good
  • grammar: excellent

Author:  Jose Gaite  on 2022-07-14  [id 2659]

(in reply to Report 2 on 2022-07-11)

First of all, I have to insist that the motivation for my study is to achieve a simple non-perturbative approach to renormalization, and that I study the best analytical equation that does not introduce parameters apart from the bare an renormalized masses and coupling constants.

It is unclear to Reviewer 2 "why one should be interested in the approximate study of Eqs. (3) or (5)." Eq (3) is the Wegner-Houghton ERG equation for the effective potential, and Eq (5) comes from its integral form (neat derivations of Eq (3) are given by Hasenfratz and Hasenfratz [4] and by Morris [15]). There have been many people interested in Eq (3), in particular, Parola, Pini, and Reatto [35], a reference that I have introduced to comply with Reviewer 2's preceding report.

However, Reviewer 2 seems to have two concerns: the sharp cutoff in the Wegner-Houghton ERG equation and the restriction to the effective potential, which misses the terms with field derivatives (these are related concerns, as the sharp cutoff may prevent a generalization that includes these terms). As in his/her preceding report, Reviewer 2 is concerned about problems that do not pertain to my manuscript. In it, the sharp cutoff is convenient, because it gives rise to simpler renormalization equations and allows us to make simpler contact with perturbation theory. On the other hand, the effective potential is sufficient to study symmetry breaking patterns and broken symmetry phases with some generality, and this is my next goal. I may next consider studying the generalization to the terms with field derivatives, and for this, my idea is to rely on the relationship with the Dyson-Schwinger equations.

Reviewer 2 now considers that a sentence in the Discussion is "highly misleading"; namely, the sentence: "The necessity of a boundary condition at infinity [...] our simplified approach constitutes a step forward towards successful methods of integration of the exact equation." This sentence was already present in the first submitted version but Reviewer 2 did not complain in his/her preceding report.

Reviewer 2 says that there are no major difficulties in numerical integrations and that "the boundary at infinity [...] can also be dealt with by compactifying the field". I disagree on several respects.

Morris [15] certainly proved that a boundary condition at infinity is needed. In regard to numerical integrations, I only say: "Arguably, the absence of such boundary conditions has hampered attempts at numerical integration". This I know to be true in some cases, at least. In the revised version, I have added a pertinent comment: "a rather extensive study of the relation between the various boundary conditions [...] should be carried out". Presumably, it is a reasonable request. Perhaps, such study has already been carried out? Saying that "there are no major difficulties" should not be enough, in my opinion.

To resort to compactifying the field is wrong, because this compactification is going to change the effective potential and the theory. This has been shown for the scalar field theory in two dimensions. The compactification of just the Gaussian field theory gives rise to distinguishable critical theories, according to the compactification radius (see Ch. 9 of Vol. 2 of Statistical Field Theory, C. Itzykson and J.-M. Drouffe, CUP 1989, especially Fig. 6). Most certainly, the same happens in less trivial models and in higher dimensions.

However, if Reviewer 2 thinks that my sentence assumes too much generality, I can replace "has hampered attempts" by "has hampered some attempts". Or I would like to know how Reviewer 2 would phrase it.

Anonymous on 2022-07-19  [id 2669]

(in reply to Jose Gaite on 2022-07-14 [id 2659])
Category:
reply to objection

I will only briefly reply to this comment, as I am still convinced that this manuscript should not be published in SciPost.

  • Maybe it is a sign that the relevant literature about the Wegner-Houghton ERG equation is 30 years old: people have pretty much stopped to use it because it is very hard to go beyond the LPA. The LPA has been studied at length over the past 40 years, and the relevant literature on the ERG is now focused on more advanced approximation. I still do not see what is gained from this manuscript.

  • I only picked up on the boundary condition at infinity in my second report because the author has listed it as an insight from his approach. I still consider it is not the case. One can solve the flow of the LPA numerically on a grid, without much trouble. It is such the case that the treatment of the LPA (with arbitrary regulator) is left to an Appendix in the recent review Physics Reports 910, 1 (2021), and the numerical integration of the flow is shown in Fig. B-31. We can note in passing that the solution in the broken symmetry phase is also shown (a "future work" for the author). Also, I might not have been clear enough when talking about compactification. I just meant that if one is interested in including the large field limit when solving the flow equation (which is not really necessary as noted above), one can use a different variable, say x= phi^2/(1+phi^2).

Also, I did not ask to cite Parola et al. but Phys. Rev. E 94, 042136 (2016).

Author:  Jose Gaite  on 2022-07-21  [id 2673]

(in reply to Anonymous Comment on 2022-07-19 [id 2669])

My work is not intended to detract from numerical integration approaches, but to complement them and provide a new perspective, through a simple analytical approach to non-perturbative renormalization. I am certain that there are researchers who appreciate my work.

I thank the reviewer for clarifying that he/she meant that one can use a change of variable that places infinity at a finite point. This is useful for a numerical solution, but it is also the standard way to study asymptotic behavior in the theory of differential equations. In fact, that is what I do in my paper, in Sect 5, with the change of variable y=1/x (this is the standard change of variable). But you have to study the behavior of the differential equation at the point y=0, as I do, and I find that it is a singular point. I doubt that a numerical integration on a grid can achieve this goal. Since the asymptotic behavior of the exact equation is very difficult to study, my analytical study gives insight into that behavior.

I have referred to two articles that study broken symmetry phases with the ERG. Sorry for not citing PRE 94, 042136 (2016) and in its place citing Parola et al: in my review of the subject, I found it to be the basic reference.

---

## Round 3 · Referee Report · Anonymous (Referee 3) · 2022-8-3

Strengths

1 - Well written.
2 - Thorough comparison with relevant literature.
3 - Explicit computations.
4 - Potential for future directions.

Report

The author has addressed several weaknesses pointed out in my last report, and has now included clearer explanation of the main equation under study, as well as clearer discussion on future directions not addressed in this work.

The largest issue that I pointed out in my previous report was the lack of motivation for analyzing this approximation to the effective potential. While I still believe that this motivation could be questioned, I appreciate the author’s point that there could be interest in studying such an approximation if it leads to analytic insights, and that the author has taken steps to provide evidence for such insights. In particular, the detailed, quantitative comparison made by the author between the results of their approximation and the previous literature suggests further inspection of these methods, and potential for follow-up work. As such, I agree that the paper meets the criterion that it “Opens a new pathway in an existing or a new research direction, with clear potential for multipronged follow-up work”. I recommend publication of the paper.

---

## Round 3 · Author Response

Reply to Reviewer 1:

I thank the reviewer for appreciating the strengths of my paper. I respond to the criticisms to the extent that it is possible and I explain the changes that I have made.

Shepard et al's indeed present equation (2) as an improvement of the one-loop approximation, but they do not consider the integral formulation of the ERG for the effective potential and, hence, they do not write equation (5). This equation constitutes an exact formulation of the ERG, but needs to be approximated, either analytically or numerically. I argue that my analytical approach is the best one that does not introduce new functions (or parameters). This is further demonstrated by the example of mass renormalization in Sect. III.A.

Nevertheless, one can ask, as Reviewer 3 does, for the error in my approximation. I have introduced two new paragraphs with comments, in Sect. II, to explain how the error in my approach compares to the error in numerical approximations.

The symmetry breaking by radiative corrections (quantum fluctuation effects) found by Coleman and Weinberg in 4-dimensional scalar electrodynamics is somehow the opposite situation to the one found in the theory of a single scalar field in three dimensions. Indeed, in the latter, the $\phi$-reflection symmetry that is spontaneously broken in the classical potential for m_0^2 < 0 is recovered by fluctuation effects, since m^2 > 0 as long as m_0^2 does not get too negative.

Coleman and Weinberg's finding, studied further in an abundant literature, can surely be studied non-perturbatively with similar methods to the ones I use for a single scalar field, but this is beyond the scope of the present work. I have added a paragraph about symmetry breaking in the Introduction.

There are several issues about symmetry breaking with which one can deal; for example, symmetry breaking with several fields, including fermionic fields, and breaking of various discrete or continuous symmetries. This would require further research and is beyond the scope of the present paper.

I have added five new articles in the bibliography.

Reply to Reviewer 2:

I thank the reviewer for reading the paper and suggesting references that will be useful in my future work. As I understand the review, it contains two criticisms.

The first criticism concerns the motivation for studying an approximation to the LPA flow equation instead of just this equation. I have shown that my approach is the best analytical approximation to the ERG equation for the potential that does not introduce new functions (or parameters) apart from the classical and renormalized potentials. Of course, various numerical approximations to the equation have been devised. The motivation for my approach is to have the insight into non-perturbative renormalization that numerical integrations do not provide. I have introduced in Sect. II two new paragraphs with comments in this regard.

Let me summarize the insights that we obtain with my approach: - Proper treatment of the boundary conditions for large values of the field. - Derived from it, in combination with the particular solution in 3d of my Sect. 4.A, the focus is put on a sextic field theory and on tricritical behavior. - Simple connection with perturbative renormalization. - Simple connection with another non-perturbative approach, namely, the Dyson-Schwinger equations. This is important, since there are computer packages that numerically solve both the ERG equations and the Dyson-Schwinger equations but provide no physical insight.

The second criticism is that my equation "is not able to capture some phenomena that are very well understood in the context of ERG". I gather that these phenomena are convexity of the effective potential and its non-analyticity at zero field for zero mass. It puzzles me how the reviewer has come to such conclusion, since I study the non-analyticity at zero field for zero mass at the beginning of Sect. 4, and I insist later, e.g., in page 21, that "when we approach this value [m=0] ... the solution of Eq. (11) is not analytic at x=0".

As concerns the convexity of the effective potential in symmetry broken phases, I admit that the sentence about symmetry broken phases in the last section was badly expressed. However, it mainly stated that my approach needs "further elaboration" to study such phases. I am happy to know of very relevant references in this regard, but I think that I cannot be blamed for not having searched in the literature a question that I had left for the future.

I have added a new paragraph about symmetry breaking in the Introduction, referring to the ERG study of the effective potential in symmetry broken phases, and I have also changed the last paragraph of the Discussion. I will study thoroughly this theme for my next work.

In total, I have added five new articles in the bibliography.

Reply to Reviewer 3:

I thank the reviewer for appreciating the strengths of my paper. I respond to the criticisms to the extent that it is possible and I explain the changes that I have made.

I am not certain that I understand why the reviewer questions the motivation for my study. The motivation for studying non-perturbative renormalization is clear, as it clarifies and improves on the results of perturbative renormalization. The reviewer may have misunderstood the notation U_DS. It is not a special type of potential but it is the standard effective potential, which accounts for the effect of fluctuations (either quantum or thermal). It allows us to calculate the renormalized coupling constants. The subscript DS is put to highlight the connection of its formulation in terms of integral equations with the Dyson-Schwinger equations. I have made some changes in page 6 to clarify this point.

The error when going from equation (5) to Shepard et al's equation (2) is not pertinent to my approach, which is more elaborate. However, I have to admit that quantifying the error of my approach seems to be a difficult task. One can derive inequalities between renormalized and bare parameters, such as m^2 > m_0^2. However, I do not see how to obtain two-sided bounds for couplings constants, and much less do I see how to obtain uniform bounds for approximations of the effective potential. As this rather mathematical question needs no urgent solution, in my opinion, I have limited myself to include two new paragraphs with comments, in Sect. II.

The sentence about symmetry broken phases in the last section was badly expressed and had ignored some work that has been pointed out by Reviewer 2. I have added a paragraph about symmetry breaking in the Introduction, referring to that work, and with additional remarks. I have also changed the last paragraph of the Discussion.

In total, I have added five new articles in the bibliography.

---

## Round 3 · List of Changes

Summary of changes:

  • New paragraph about symmetry breaking in the Introduction.

  • Changes in page 6 to clarify the name and role of U_DS.

  • Two new paragraphs in Sect. II with comments on comparison with numerical approaches and error estimation.

  • Changes in the last paragraph of the Discussion.

  • Five new articles in the bibliography.

---

## Editorial Decision

published